# Mitigating Test-Time Bias for Fair Image Retrieval

**Fanjie Kong**
Duke University
fanjie.kong@duke.edu

**Shuai Yuan**
Duke University
shuai@cs.duke.edu

**Weituo Hao**
TikTok Inc.
weituohao@tiktok.com

**Ricardo Henao**
Duke University
KAUST
ricardo.henao@duke.edu

## Abstract

We address the challenge of generating fair and unbiased image retrieval results given neutral textual queries (with no explicit gender or race connotations), while maintaining the utility (performance) of the underlying vision-language (VL) model. Previous methods aim to disentangle learned representations of images and text queries from gender and racial characteristics. However, we show these are inadequate at alleviating bias for the desired equal representation result, as there usually exists test-time bias in the target retrieval set. So motivated, we introduce a straightforward technique, Post-hoc Bias Mitigation (PBM), that post-processes the outputs from the pre-trained vision-language model. We evaluate our algorithm on real-world image search datasets, Occupation 1 and 2, as well as two large-scale image-text datasets, MS-COCO and Flickr30k. Our approach achieves the lowest bias, compared with various existing bias-mitigation methods, in text-based image retrieval result while maintaining satisfactory retrieval performance. The source code is publicly available at `https://github.com/timqqt/Fair_Text_based_Image_Retrieval`.

## 1  Introduction

Image search on the web based on text-based image retrieval (TBIR) (Lew et al., 2006) involves interpreting a user's (text) query and returning corresponding images that are considered relevant in terms of semantic meaning (Chen et al., 2015). With recent advancements in multi-modal representation learning, Vision-Language (VL) models such as CLIP have been widely used to enhance the efficacy of text-based image retrieval (Radford et al., 2021; Cao et al., 2022). These models are usually trained on vast datasets that consist of millions of text-image pairs scrapped from the web, which inevitably manifest societal biases especially for neutral queries (Wang et al., 2021a; Hall et al., 2023; Wang et al., 2021b), *i.e.*, queries without explicit demographic (gender or race) connotations. In Figure 1, we show image retrieval results for the neutral query "Bus Driver" using CLIP and an unbiased alternative delivered by the proposed approach.

In our work, we adhere to *equal representation* as our fairness objective (Mehrotra and Celis, 2021; Kay et al., 2015), as an alternative to *proportional representation* (Jalal et al., 2021; Berg et al., 2022). The latter, which aligns the demographic proportions in the retrieval results with those in the dataset, which is susceptible to biases during collection (Kay et al., 2015). Instead, equal representation for all demographic groups of interest attempts to mitigate (obscure) the influence of any inherent biases.

Previous works have been dedicated to developing unbiased VL models that promote fairness in TBIR tasks (Wang et al., 2021a; Berg et al., 2022; Wang et al., 2022; Kim et al., 2023; Chuang

37th Conference on Neural Information Processing Systems (NeurIPS 2023).

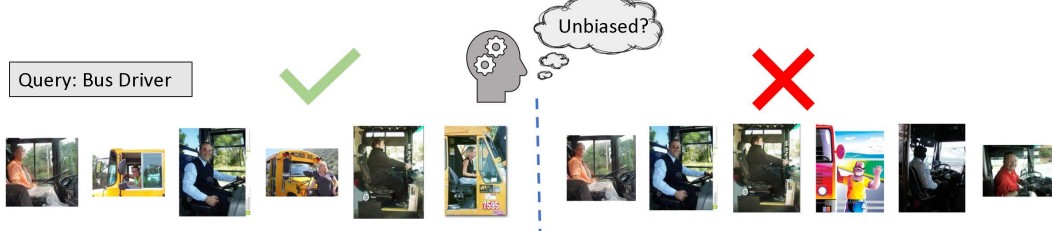

Figure 1: Text-based image retrieval (TBIR) results of a neutral query. Right: all returned images are male bus drivers, possibly pushing the message that only men (can) perform this job. Left: desirable unbiased result with equal gender representation obtained by the proposed PBM approach.

et al., 2023; Parraga et al., 2022). Berg et al. (2022) leveraged adversarial training for debiasing in which a learnable prompt is prepended to the input text queries, while an adversary seeks to discourage the image and text encoders from capturing gender or race information for retrieval. Alternatively, Wang et al. (2021a) discarded the components of the visual and text representations that are highly correlated with gender, by estimating the mutual information between these components and gender attributes. Further, Wang et al. (2022) neutralized the gender contributions in image representations by enforcing equal contributions from gender (male and female) features using a bias contrast loss, while maximizing the contribution of gender-irrelevant features. These studies largely rely on demographic attributes that can be somewhat visually perceived. For instance, gender attributes are derived from perceived characteristics of masculinity or femininity, leading to labels such as "Male" or "Female". Similarly, race-related attributes are often characterized based on skin tones and typically categorized into groups such as "Fair Skin" and "Dark Skin" (Kay et al., 2015; Celis and Keswani, 2020). However, these methods primarily focus on debiasing the gender or race encoding within VL models, which may be insufficient due to bias or imbalance in the image retrieval candidate pool (test set). In contrast, though we also mitigate bias in TBIR using the same demographic attribute annotations as other works, the proposed solution is simpler in the sense that it forgoes the need for access to gradients or retraining the VL model.

In this paper, we start Section 3.1 by defining the text-based image retrieval task in the context of a fairness objective focused on equal representation. We then analyze the effectiveness of existing bias reduction methods in Section 3.4 and demonstrate how they fall short in achieving equal representation, mainly due to imbalances in the test-time image retrieval set. Based on this observation, in Section 3.5 we propose a simple yet effective post-processing debiasing method called post-hoc bias mitigation (PBM) that creates fair retrieval subsets guided by predicted gender (or race) attributes obtained from either an off-the-shelf gender (or race) classifier or zero-shot inference using a pre-trained VL model. In Section 4, we evaluate PBM on real-world web image search (Kay et al., 2015; Celis and Keswani, 2020) and large-scale image-text datasets such as MS-COCO (Chen et al., 2015) and Flickr30K (Plummer et al., 2015). By comparing our approach to various existing bias-mitigation techniques, we show that our method achieves the lowest bias while maintaining satisfactory retrieval performance, as evidenced by both quantitative and qualitative results. Importantly, PBM strives to provide a more unbiased, fair and diverse visual representation of different groups in search results, ultimately promoting social welfare by challenging existing social perceptions of gender and race.

The summarized contributions of this work are:

- We present an analysis of the effect of existing debiasing methods on VL models, and highlight their insufficiency in achieving equal representation for text-based image retrieval.

- We propose PBM, a straightforward and efficient test-time post-processing debiasing algorithm that generates fair retrieval subsets, guided by predicted gender/race information obtained from an off-the-shelf classifier or inferred via zero-shot using the VL model itself.

- We evaluate PBM on two real-world web image search and two large-scale image-text datasets for text-based image retrieval, and compare with existing bias-mitigation techniques, demonstrating its effectiveness in achieving the lowest bias among all tested methods.

## 2   Related work

**Text-based image retrieval**   Text-based image retrieval is the process of searching and retrieving images from a large database using textual descriptions or keywords as queries. The task is usually tackled by image-text feature alignment (Cao et al., 2022), which embeds image and text inputs into a shared feature space such that relevant image and text features are close to each other. In recent years, substantial progress has been made due to the emergence of large-scale datasets, as well as the development of effective deep learning-based image-text models. Pioneering works include DeViSE (Frome et al., 2013), which bridges the *semantic gap* between image content and textual descriptors by aligning CNN-based features of images with textual embeddings from ImageNet labels. Subsequent approaches, such as VSE++ (Faghri et al., 2017) and SCAN (Lee et al., 2018), further refine these joint embeddings to improve retrieval performance. Recently, OpenAI's CLIP (Radford et al., 2021) has emerged as a powerful approach for image retrieval based on similarity matching between extracted image and text features. CLIP leverages a pre-trained transformer model, jointly optimized for both image and text understanding, which allows it to effectively match images and textual descriptions.

**Fairness in machine learning**   Recent studies have highlighted numerous unfair behaviors in machine learning models (Angwin et al., 2016; Buolamwini and Gebru, 2018). For example, a risk assessment algorithm used in the United States criminal justice system predicted that Black defendants were more likely to commit future crimes than white defendants, even after controlling for criminal record (Angwin et al., 2016). Moreover, individuals with different gender and skin-tones are likely to receive disparate treatment in commercial classification systems such as Face++, Microsoft, and IBM systems (Buolamwini and Gebru, 2018). Consequently, there has been a surge in demand and interest for developing methods to mitigate bias, such as regularizing disparate impact (Zafar et al., 2015) and disparate treatment (Hardt et al., 2016), promoting fairness through causal inference (Kusner et al., 2017), and incorporating fairness guarantees in recommendations and information retrieval (Beutel et al., 2019; Morik et al., 2020).

**Fairness in vision-language models**   After some studies revealed the bias of using VL models in downstream tasks (Wang et al., 2021a; Hall et al., 2023; Wang et al., 2021b), efforts to address and mitigate these biases in VL models have gained increasing attention. Existing solutions for fair vision-language models can be generally classified into pre-processing, in-processing, and post-processing methods. Pre-processing techniques usually involve re-weighting or adjusting the training data to counter imbalances across demographic attributes, while preserving the utility of the dataset for the target task (Friedler et al., 2014; Calmon et al., 2017). In-processing methods focus on altering the training objective by incorporating fairness constraints, regularization terms or leveraging adversarial learning to obtain representations invariant to gender/race (Berg et al., 2022; Wang et al., 2023; Xu et al., 2021; Cotter et al., 2019). Post-processing approaches achieve fairness by applying *post-hoc* corrections to a (pre-)trained model (Cheng et al., 2021; Calmon et al., 2017) or via feature clipping (Wang et al., 2021a) on the output of image-text encoders based on mutual information.

Our work lies in the post-processing category of debiasing methods that encourages equal representation of diverse demographics. We also identified the fair subset selection approach used in Mehrotra and Celis (2021) as a potential post-hoc debiasing method for TBIR. While Mehrotra and Celis (2021) shares our goal of ensuring equality of gender/race attributes in the set of results, their focus did not extend to the TBIR scenario with an underlying VL model nor detail an effective method for obtaining accurate demographic attributes for debiasing. More importantly, they assumed demographic attributes seen by their algorithm to be available ground truth labels subject to noise. This assumption creates difficulties when attempting to adapt their method to a real-world problem such as TBIR. Complementary, our approach is meant to address these deficiencies by providing a practical debiasing procedure, that includes acquiring demographic attributes.

**Gender/Racial Bias in Web Image Search**   Our research is closely related to studies in the Human Computer Interaction community that demonstrated gender inequality issues in current online image search systems (Kay et al., 2015; Noble, 2018; Celis and Keswani, 2020). These studies revealed how gender bias in occupational image search results influences people's perceptions about the presence of men and women in various professions. Our work builds upon these findings by examining the gender and racial bias in image search algorithm and offering innovative solutions for reducing bias in popular image retrieval framework using pre-trained VL model, such as CLIP (Radford et al., 2021).

## 3 Method

### 3.1 Problem formulation

**Image retrieval** Suppose $C$ is the set of all text queries of interest (gender-neutral defined below), and $\mathcal{V} = \{v_i\}_{i=1}^N$ is a database of $N$ images, from which we retrieve images given query inputs. Each query input $c \in C$ is relevant to a ground-truth set of images $V_c^* \subseteq \mathcal{V}$ provided by human annotators.

The text-based image retrieval task aims at finding images from the database so they best match the text query inputs. Specifically, given any text query $c \in C$ and a fixed retrieval size $K$ ($K \ll N$) as inputs, the goal is to design an algorithm that returns a bag of $K$ images $V_{c,K} \subseteq \mathcal{V}$, where $|V_{c,K}| = K$, such that $V_{c,K}$ contains as many relevant images from $V_c^*$ as possible.

For evaluation purposes, a top-$K$ recall score ("Recall@$K$") is usually computed to quantify whether the most relevant images are included in the retrieval output. Specifically, we write

$$\text{Recall@}K = \frac{\sum_{c \in C} |V_{c,K} \cap V_c^*|}{\sum_{c \in C} |V_c^*|} \times 100\%,$$

where $K$ is selected specifically for each dataset so that $|V_c^*| \leq K$ for most queries $c \in C$; this way, the recall can be realistically close to 100%.

**Debiasing in image retrieval** Using gender as a motivating scenario, we are interested in gender debiasing in cases where queries relate to human characteristics with no gender connotations, which we refer to as *gender-neutral queries*. For example, queries could be occupations that are widely open to all individuals, irrespective of gender (Organization, 2019), such as chef, nurse, social worker, *etc.*

We anticipate that gender-neutral queries should yield image retrieval sets that comprise equal representation of both male and female associated images, *i.e.*, in a 1:1 ratio, which is consistent with the equal representation goal discussed by Wang et al. (2021a, 2023); Mehrotra and Celis (2021); Mehrotra and Vishnoi (2022). Note that we use gender debiasing as an example, but the same setting can be generalized to other debiasing issues such as racial discrimination, as we will demonstrate in the experiments.

Specifically, assume each image $v \in \mathcal{V}$ has a gender attribute $g(v) \in \{+1, -1\}$, which corresponds to the two genders manifested in the image, male and female, respectively. For each gender-neutral query $c$, the gender bias of the resulting retrieved bag of images $V_{c,K}$ is defined as the normalized absolute difference between the numbers of images of each gender, *i.e.*,

$$B(V_{c,K}) = \frac{1}{K} \left| \sum_{v \in V_{c,K}} \mathbb{1}\{g(v) = +1\} - \sum_{v \in V_{c,K}} \mathbb{1}\{g(v) = -1\} \right| = \frac{1}{K} \left| \sum_{v \in V_{c,K}} g(v) \right|, \tag{1}$$

which ranges from $\frac{1}{K}\text{mod}(K, 2)$ (minimum bias) to 1 (maximum bias). In our retrieval approach, we average $B(V_{c,K})$ across all gender-neutral queries $c \in C$ as an evaluation metric for fairness, *i.e.*,

$$\text{AbsBias@}K = \frac{1}{|C|} \sum_{c \in C} B(V_{c,K}) = \frac{1}{|C|} \sum_{c \in C} \frac{1}{K} \left| \sum_{v \in V_K^c} g(v) \right|.$$

The goal is to minimize AbsBias@$K$ while maintaining a satisfactory retrieval Recall@$K$.

### 3.2 Similarity-based image-text matching

**Overall framework** As in previous works (Singh et al., 2003; Bai et al., 2014; Zaidi et al., 2019; Cao et al., 2022; Mukhoti et al., 2022), we use a similarity matching approach to tackle the image retrieval task. Such an approach involves aligning image and text features from largely pre-trained vision and language models during training time. At inference time, we rank images based on whether their features are similar to the input query text features and use the top-ranked images as our retrieval output. Specifically, we use an image encoder network $f_\phi(\cdot)$ and a text encoder network $f_\psi(\cdot)$ to embed both image $v$ and text $c$ inputs into a shared $d$-dimensional feature space as $f_\phi(v), f_\psi(c) \in \mathbb{R}^d$. A cosine similarity score $S(v, c)$ is then computed to quantify the relevance between $v$ and $c$ (Singh et al., 2003; Bai et al., 2014; Wang et al., 2021a).

**Training**    As in previous works (Chen et al., 2020; Radford et al., 2021), the training of the image and text encoders is achieved through optimizing an NT-Xent (Normalized Temperature-scaled Cross Entropy) loss with stochastic gradient descent on mini-batches.

**Inference**    Given a new image database $\mathcal{V}^{\text{test}}$, for each new query input $c \in C^{\text{test}}$, we compute its similarity score $S(v_i, c)$ with every image $v_i \in \mathcal{V}^{\text{test}}$ and then pick the top-$K$ images that have the maximum similarity scores to form our retrieved set $V_{c,K}$.

### 3.3    Fairness criterion for image retrieval

Fairness can be defined in numerous ways in favor of different situation (Saxena et al., 2020). In the context of fair image retrieval (Wang et al., 2021a; Berg et al., 2022), we define our fairness objective as *equal representation*, implying that the retrieved image set through the retrieval algorithm should encompass an equal number of samples from each demographic group. We formally define our criterion as follows:

**Definition 3.3.1 (Equal Representation)** *An image retrieval algorithm satisfies equal representation with respect to binary demographic attributes $g(v)$ if,*

$$\mathbb{E}_{V_c \sim P}[\mathbb{E}_v[\mathbb{1}(g(v) = +1)]] = \mathbb{E}_{V_c \sim P}[\mathbb{E}_v[\mathbb{1}(g(v) = -1)]]$$

*where $P$ represents the distribution of retrieved image set $V_c$ corresponding to neutral queries c.*

The above definition implies $\mathbb{E}[B(V_c)]_{V_c \sim P} = 0$, *i.e.*, the expected bias is zero for any retrieved image sets. Our definition of *equal representation* is equivalent to "Equal opportunity for ranking distributions" introduced by Singh and Joachims (2017) when ranking position bias is constant.

### 3.4    Bias analysis

Using the image retrieval framework defined above as a basis, it is important to incorporate strategies to reduce gender (and race) biases of the retrieval output $B(V_{c,K})$. Below, we briefly analyze existing methods and discuss their limitations.

**Existing methods**    Most methods address the gender fairness issue by enforcing model features to be less dependent on gender information. This is achieved by mainly two types of approaches, namely adversarial training (Edwards and Storkey, 2015; Berg et al., 2022; Xu et al., 2021) and mutual information (MI) minimization (Wang et al., 2021a, 2023). Specifically, adversarial training involves training a separate (adversarial) classifier network by adding an adversarial loss so that the adversarial network cannot distinguish gender given the encoded image features (Edwards and Storkey, 2015; Berg et al., 2022; Xu et al., 2021). Alternatively, MI minimization aims at reducing the MI between feature distribution and their gender by clipping feature dimensions highly correlated with gender (Wang et al., 2021a). Both methods encourage the model to extract features that are *independent* of gender. However, we argue below that enforcing this kind of independence between image features and gender is not sufficient to effectively eliminate gender bias in image retrieval.

**Illustrative example**    We start by showing debiasing results using mutual information minimization. Similar insights can be obtained via adversarial training, which is shown in the Supplementary Material (SM). Figure 2 shows an example for the query "engineer" comparing three methods, namely, CLIP (no debiasing), MI-based debiasing and the proposed PBM (described below). For each method, we first compute, sort and group the similarity scores of all images into 1% quantiles ($\sim$ 30 images each). Then, for each quantile, we compute the gender bias as defined in (1). This analysis shows the relationship between gender bias and similarity scores. Note that for retrieval, we use samples with the largest similarity scores to form our retrieved bag of images, thus only the right-most data points (highlighted in the figure) are part of the retrieval set. We show percentiles in the figure to emphasize similarity rank rather than the similarity scores *per se*.

As shown in Figure 2(a), the original image retrieval algorithm with no debiasing based on CLIP tends to assign higher scores (at larger percentiles) to samples associated with male attributes, as evidenced by the high correlation between gender and the similarity scores. This makes the final retrieval output largely biased towards male samples. In contrast, Figure 2(b) shows the result of debiasing via MI minimization  (Wang et al., 2021a). Using the regression line as a visual guide, we

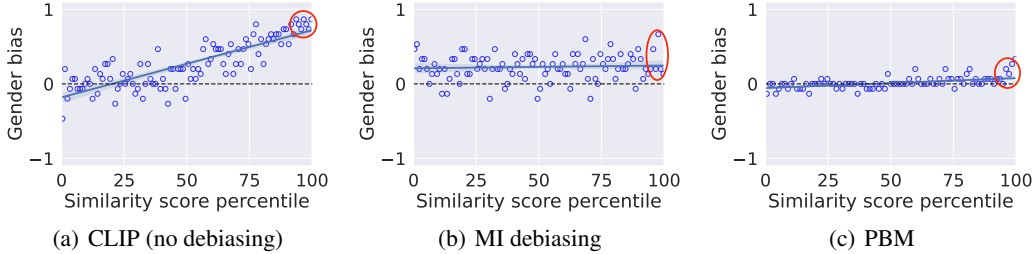

(a) CLIP (no debiasing)  (b) MI debiasing  (c) PBM

Figure 2: Gender bias distribution for different methods using "engineer" as query. We compute similarity scores for all images from the test image set and plot them against gender bias in 1% quantile increments. The red circle marks the top-$K$ window covering the final retrieval output $V_{c,K}$.

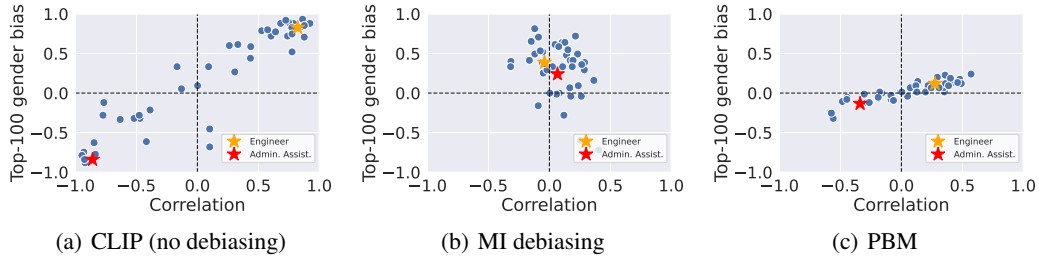

(a) CLIP (no debiasing)  (b) MI debiasing  (c) PBM

Figure 3: Comparing top-100 retrieval gender bias with full set similarity gender bias Spearman's correlation. For each query (occupation), we visualize the correlation between similarity score and gender bias against the top-100 retrieval gender bias. Two typical examples: "engineer" and "administrative assistant" are highlighted for illustration.

see that the correlation between gender and similarity score is close to zero. However, the gender bias is consistently larger than zero across the range of similarity scores, so the final retrieval is still biased. A desirable debiasing outcome is that for which the regression line aligns with the dotted line (zero gender bias across the similarity range), as shown in Figure 2(c). This is accomplished by the proposed PBM (described below).

**Example with multiple queries** In order to get a more generalizable understanding of the behavior shown in Figure 2, we repeat the analysis with all 45 occupations from the dataset provided by Kay et al. (2015) and visualize the results in Figure 3. Specifically, for each query (occupation), we calculate the Spearman's rank correlation between all (test-set) similarity scores and gender bias similar to Figure 2. We use Spearman's rank correlation because it is more appropriate to quantify associations between (variable) rankings. Moreover, we also calculate the gender bias of the final retrieval bag ($K = 100$) for each occupation. Consistent with the illustrative findings on the "engineer" query, in Figure 3(a), we see a large correlation between similarity scores and gender bias as well as a large retrieval-set gender bias. MI minimization in Figure 3(b) manages to reduce the correlation between similarity score and gender but suffers from considerable gender bias; mostly clustered around 0.4. The proposed PBM not only reduces the correlation between similarity scores and gender bias but also significantly pushes the bias of each retrieval result to zero, which is the sought target debiasing outcome.

**Insights** Our observations can be explained and understood from a theoretical perspective. Previous methods are mostly concerned with making their image features independent of gender information. For a fixed query $c$, suppose we sample a random image from the test image data distribution $v \sim \mathcal{V}^{\text{test}}$, its feature distribution $f_\phi(v)$ is independent of gender distribution $g(v)$. Since the query feature $f_\psi(c)$ is constant, the similarity score $S(v, c)$, defined in Section 3.2, is a deterministic function of $f_\phi(v)$ and is thus also independent of gender $g(v)$. This is why the results for MI-based unbiasing shown in Figure 2(b) and 3(b) show little correlation between similarity scores and gender bias. Similar results for debiasing with adversarial learning can be found in the SM.

However, as shown in Figure 2(b), there is still a consistent gender bias across similarity score values. This means that the gender distribution in each window tends to manifest the gender distribution of

---

**Algorithm 1** Post-hoc Bias Mitigation (PBM).

---

**Input**: Text query $c$, retrieval size $K$, image database $\mathcal{V}^{\text{test}}$, similarity measure from pre-trained vision-language models $S(\cdot, \cdot)$, and gender prediction model $\hat{g}(\cdot)$.
**Output**: Image retrieval bag $V_{c,K}$.

1: Split $\mathcal{V}^{\text{test}}$ into $\mathcal{V}^{\text{test}}_{+1}$, $\mathcal{V}^{\text{test}}_{-1}$ and $\mathcal{V}^{\text{test}}_{\text{N/A}}$ using the gender prediction model $\hat{g}(\cdot)$;
2: Let $V_{c,K} = \varnothing$;
3: **while** $|V_{c,K}| < K$ **do**
4: $\quad v_{+1} = \arg\max\limits_{v \in \mathcal{V}^{\text{test}}_{+1}} S(v, c); \quad v_{-1} = \arg\max\limits_{v \in \mathcal{V}^{\text{test}}_{-1}} S(v, c); \quad v_{\text{N/A}} = \arg\max\limits_{v \in \mathcal{V}^{\text{test}}_{\text{N/A}}} S(v, c);$
5: $\quad$ **if** $[S(v_{+1}, c) + S(v_{-1}, c)] / 2 > S(v_{\text{N/A}}, c)$ **then**
6: $\quad\quad V_{c,K} \leftarrow V_{c,K} \cup \{v_{+1}, v_{-1}\}; \quad \mathcal{V}^{\text{test}}_{+1} \leftarrow \mathcal{V}^{\text{test}}_{+1} \setminus \{v_{+1}\}; \quad \mathcal{V}^{\text{test}}_{-1} \leftarrow \mathcal{V}^{\text{test}}_{-1} \setminus \{v_{-1}\};$
7: $\quad$ **else**
8: $\quad\quad V_{c,K} \leftarrow V_{c,K} \cup \{v_{\text{N/A}}\}; \quad \mathcal{V}^{\text{test}}_{\text{N/A}} \leftarrow \mathcal{V}^{\text{test}}_{\text{N/A}} \setminus \{v_{\text{N/A}}\};$
9: $\quad$ **end if**
10: **end while**
11: **return** $V_{c,K}$

---

the whole test image set $\mathcal{V}^{\text{test}}$, which may not be balanced. This is also confirmed in Figure 3(b), as the final output bias of most occupations is clustered around 0.4, which is precisely the gender bias of the entire image set $\mathcal{V}^{\text{test}}$. Therefore, ensuring independence, quantified here in terms of correlation, between image features and gender may not guarantee zero gender bias if the test image set itself is biased due to imbalance.

**Two types of bias** From the examples above, we can differentiate two types of bias, namely, the model bias from training and the bias from the test-time image distribution. The former can be quantified based on the correlation between similarity score and gender and only depends on the training data distribution and the way in which the model is trained. This has been previously described and studied (Caliskan et al., 2017; Zhao et al., 2017). In comparison, the latter type of bias manifests in the test phase because the test image set (the database from which we retrieve images) does not necessarily have commensurate numbers of male and female samples. We refer this type of bias as test-time bias. In fact, such proportions are usually unknown for image databases.

These two sources of bias coexist and can be addressed separately during model training and inference. Previous methods (Edwards and Storkey, 2015; Wang et al., 2021a; Berg et al., 2022) have been fairly successful at addressing the first type of bias on the training side, but neglect the test-time bias that is specific to the test set. To tackle the test-time bias, it is necessary to find a strategy targeting the test image set, so that the retrieved image genders are balanced despite the gender imbalance in the training (source) set. This very insight motivates our approach called PBM, which we describe below. As previously shown in Figures 2(c) and 3(c), PBM achieves substantially smaller retrieval gender bias than MI debiasing. Other existing approaches will be considered in the experiments.

### 3.5 Post-hoc Bias Mitigation (PBM)

To address the second type of bias induced by the imbalanced image test set, one simple idea is sub-sampling. Specifically, we could first sub-sample the image set to make sure its gender ratio is balanced *before* doing retrieval. However, a clear limitation of such an approach is that some highly relevant images may be dropped during sub-sampling, which may negatively affect retrieval quality. This problem is especially exacerbated if the test set has a very large gender bias. Alternatively, our intuition is to sub-sample *after* computing and ranking similarity scores of all images using a post-hoc method to control gender bias while sampling from the image source set, which we call Post-hoc Bias Mitigation (PBM).

A general version of the PBM algorithm, which is straightforward, is presented in Algorithm 1. For each image $v_i \in \mathcal{V}^{\text{test}}$, we first predict its gender $\hat{g}(v_i) \in \{+1, -1\}$, which splits the images into two subsets, $\mathcal{V}^{\text{test}}_{+1}$ and $\mathcal{V}^{\text{test}}_{-1}$. We then rank images from the two subsets based on their similarity scores separately. While forming the retrieval bag, we sample the top of both subsets together in pairs. Specifically, we select the top $\lfloor K/2 \rfloor$ samples from each subset. If $K$ is odd, we randomly pick one of the subsets and select one more top sample from it. This method ensures low gender bias of the retrieval output as long as gender predictions $\hat{g}(v_i)$ are accurate.

One extension of our method considers the case where gender may not be applicable to some images. For instance, our image dataset may contain cartoon characters that are not easily associated with any gender, or maybe the gender cannot be determined from the image due to body or facial coverings. In this case, we allow our gender predictions to take N/A values, yielding a $\mathcal{V}_{\text{N/A}}^{\text{test}}$ subset. Images from $\mathcal{V}_{\text{N/A}}^{\text{test}}$ do not inherently exhibit visually perceptible bias. Thus, they could be selected in favor of other male-female pairs if their similarity scores are higher and are exempt from bias measurement.

**Gender prediction**   To predict gender $\hat{g}(v)$, we consider the following two methods for different scenarios. Note that all options are considered in the experiments.

*Supervised gender classification:* For scenarios where ground-truth gender attributes are available, such as MS-COCO (Lin et al., 2014; Zhao et al., 2021), we can train a complementary small gender classification network. Using our encoded image feature $f_\phi(v)$ as input, we train a 3-layer MLP to classify gender. For datasets where gender may not be applicable to some samples, we add a N/A class to the set of labels for predictions.

*Zero-shot inference using word embeddings or prompt:* For scenarios where supervised training of gender is not possible, we can infer gender in a zero-shot manner (Radford et al., 2021; Li et al., 2022) using the implicit knowledge embedded in the text features of large pre-trained vision-language models (Radford et al., 2021; Li et al., 2022). Given that the semantics of gender and race attributes are already incorporated into the text encoder, we can extract them from word embeddings using words such as "Man" and "Woman". Alternatively, we could use prompts prepended to the occupation search query. For example, we add gender-specific adjectives like "Male" or "Female" in front of a query. Finally, we compute the (cosine) similarity score of each image $v$ with them, *i. e.*, $S(v, \text{"Male "} + c)$, $S(v, \text{"Female "} + c)$. The plus operator (+) here refers to string concatenation. The gender prediction $\hat{g}(v)$ is then generated by comparing these two similarity scores.

## 4   Experiments

We first evaluate our image retrieval algorithm on two real-world web image search datasets, Occupation 1 (Kay et al., 2015) and Occupation 2 (Celis and Keswani, 2020). We also test on two large-scale image-text datasets, MS-COCO (Lin et al., 2014) and Flickr30k (Plummer et al., 2015) to further validate the effectiveness of PBM in handling more complex text-based image retrieval scenarios.

For comparison, we consider adversarial training (Edwards and Storkey, 2015; Berg et al., 2022) and mutual information minimization (Wang et al., 2021a) as baseline methods. We also include other types of debiasing methods such as FairSample (Wang et al., 2021a), which balances the gender distribution of image-text pairs within a training batch, and FairExpec (Mehrotra and Celis, 2021), a denoised selection algorithm designed to select fair subset based on noisy demographic attributes. We use AbsBias@KK and Recall@KK as evaluation metrics as described in Section 3.1.

**Real-world web image search**   The first dataset, which we refer to as Occupation 1 (Kay et al., 2015), comprises the top 100 Google image search results for 45 gender-neutral occupation terms, such as "chef", "librarian", "primary school teacher", *etc.* Each image within this dataset is annotated with a crowd-sourced gender attribute (either "male" or "female") that characterizes the person depicted in the image. Occupation 2 (Celis and Keswani, 2020), the second dataset, includes the top 100 Google image search results for 96 occupations, where both gender and race (represented as skin-tone: fair and dark skin) annotations are provided. Notably, the gender and race attributes also include a N/A category in Occupation 2, where the annotators have chosen the option of "Not applicable" or "Cannot determine" for the gender or skin-tone represented in the image. Consequently, we treat the images labeled with N/A as neutral examples that do not contribute to the bias of retrieval, since in principle, the users cannot perceive gender or racial information from the image.

For these two datasets, we consider OpenAI's CLIP ViT-B/16 (Radford et al., 2021) as the VL model for all debiasing methods. The baselines for comparison are MI-*clip* from Wang et al. (2021a), adversarial training adapted from Edwards and Storkey (2015), and Debias Prompt from Berg et al. (2022). We have also tailored FairExpec (not originally proposed for TBIR) to our task by integrating it with CLIP and our proposed gender predictor $\hat{g}(\cdot)$. We refer to this adapted model as CLIP-FairExpec in our experiments. We test our PBM method with four variants of gender predictors $\hat{g}(\cdot)$, as discussed in Section 3.5. These variants include a supervised classifier, zero-shot inference, and ground truth from annotators. The implementation details are provided in the SM.

Table 1: Results for debiased image retrieval from Occupation 1 and 2 datasets.

| Method | Occupation 1 - Gender | | Occupation 2 - Gender | | Occupation 2 - Race | |
|---|---|---|---|---|---|---|
| | AbsBias@100 ($\downarrow$) | Recall@100($\uparrow$) | AbsBias@100($\downarrow$) | Recall@100($\uparrow$) | AbsBias@100($\downarrow$) | Recall@100($\uparrow$) |
| Random Selection | .6370 | - | .3155 | - | .4171 | - |
| CLIP Original (Radford et al., 2021) | .6231 | 58.3 | .3566 | 46.2 | .5002 | 46.2 |
| MI-*clip* (Wang et al., 2021a) | .3769 | 47.0 | .2539 | 42.2 | .4099 | 42.3 |
| Adversarial Training (Edwards and Storkey, 2015) | .2316 | 44.0 | .2603 | 37.8 | .4880 | 43.3 |
| Debias Prompt (Berg et al., 2022) | .6373 | **59.3** | .3564 | **46.2** | .4946 | **50.2** |
| CLIP-FairExpec (Mehrotra and Celis, 2021) | .2498 | 47.0 | .2619 | 44.2 | .2788 | 34.7 |
| **PBM** - Zero-shot Embedding | .0969 | 49.8 | .1150 | 42.1 | .3133 | 40.2 |
| **PBM** - Zero-shot Prompt | **.0560** | 46.1 | **.0443** | 42.5 | .2571 | 36.0 |
| **PBM** - Supervised Classifier | .1404 | 50.3 | .1171 | 42.1 | **.0955** | 37.9 |
| **PBM** - Ground-truth Gender and Skin-tone | .0000 | 49.1 | .0000 | 42.4 | .0000 | 41.3 |

Table 2: Results for debiased image retrieval from MS-COCO and Flickr30k datasets.

| Dataset | Method | Gender Bias | | | Recall | | |
|---|---|---|---|---|---|---|---|
| | | Bias@1 ($\downarrow$) | Bias@5($\downarrow$) | Bias@10($\downarrow$) | Recall@1($\uparrow$) | Recall@5($\uparrow$) | Recall@10($\uparrow$) |
| COCO-1k | SCAN (Lee et al., 2018) | .1250 | .2044 | .2506 | 47.7 | 82.0 | **91.0** |
| | FairSample (Wang et al., 2021a) | .1140 | .1951 | .2347 | **49.7** | **82.5** | 90.9 |
| | CLIP (Radford et al., 2021) | .0900 | .2024 | .2648 | 48.2 | 77.9 | 88.0 |
| | MI-*clip* (Wang et al., 2021a) | .0670 | .1541 | .2057 | 46.1 | 75.2 | 86.0 |
| | **Our PBM** | **.0402** | **.0961** | **.1082** | 37.3 | 73.6 | 84.8 |
| COCO-5k | SCAN (Lee et al., 2018) | .1379 | .2133 | .2484 | 25.4 | 54.1 | 67.8 |
| | FairSample (Wang et al., 2021a) | .1133 | .1916 | .2288 | 26.8 | **55.3** | **68.5** |
| | CLIP (Radford et al., 2021) | .0770 | .1750 | .2131 | **28.7** | 53.9 | 64.7 |
| | MI-*clip* (Wang et al., 2021a) | .0672 | .1474 | .1611 | 27.3 | 50.8 | 62.0 |
| | **Our PBM** | **.0492** | **.1006** | **.1212** | 22.3 | 50.5 | 61.9 |
| Flickr30K | SCAN (Lee et al., 2018) | .1098 | .3341 | .3960 | 41.4 | 69.9 | 79.1 |
| | FairSample (Wang et al., 2021a) | .0744 | .2699 | .3537 | 35.8 | 67.5 | 77.7 |
| | CLIP (Radford et al., 2021) | .1150 | .3150 | .3586 | **67.2** | **89.1** | **93.6** |
| | MI-*clip* (Wang et al., 2021a) | .0960 | .2746 | .2951 | 63.9 | 85.4 | 91.3 |
| | **Our PBM** | **.0360** | **.1527** | **.1640** | 41.2 | 85.3 | 92.6 |

We summarize the experimental results in Table 1. Compared with other methods, PBM variants achieve significantly lower AbsBias@100 while maintaining comparable Recall@100 scores. The male:female ratio of images in Occupation 1 is approximately 61:49, thereby mitigating the first type of bias by MI-*clip*, adversarial training, debias prompt and random selection is not enough. The CLIP-FairExpec cannot achieve better results since the FairExpec algorithm relies on the independence assumption, while the samples in our test dataset are correlated. This happens because images for occupations are collected from the same Google image search, which is biased. Further, FairExpec requires reliable probabilistic predictions for gender, however, in our setting the gender attributes are provided by an off-the-shelf MLP predictor based on visual features. In such a scenario, the label estimates yielded by the off-the-shelf MLP may not be always trustworthy, as the domain has shifted during inference. Consequently, these estimates could include misleading information, resulting in undesirable debiasing outcomes for CLIP-FairExpec. Moreover, it should be noted that Debias Prompt (Berg et al., 2022) always achieves the highest Recall@100, since we utilize their publicly accessible model, which has been fine-tuned on the Flickr30k dataset to enhance the performance of text-based image retrieval.

**Large-scale text-based image retrieval**  We consider MS-COCO (Lin et al., 2014) and Flickr30k (Plummer et al., 2015). Our setup aligns with Wang et al. (2021a), where the gender attributes are directly inferred from the text captions of images. We consider the same baseline models as Wang et al. (2021a), which are SCAN (Lee et al., 2018) and CLIP, as well as their proposed approach, FairSample and MI-*clip*. It should be noted that FairSample is specifically designed to debias SCAN, a specialized in-domain VL model. We choose the best performing PBM model, which leverages the pre-trained classifier for gender attributes.

The performance metrics in Table 2 are consistent with Wang et al. (2021a), which resemble our AbsBias@$K$ metric, with the omission of using absolute values in the sum. This bias metric is computed as Bias@$K = \frac{1}{|C|} \sum_{c \in C} \frac{1}{K} \sum_{v \in V_K^c} g(v)$. From the Table 2, we see that PBM maintains the bias to a minimum even when dealing with intricate text queries or images of complex scenes.

**Bias-performance trade-off analysis**  In Figure 4, we show the trade-off between retrieval performance and bias for MI-*clip*, adversarial training and the four PBM variants. We choose MI-*clip* and adversarial training for comparison, as these models offer implementation simplicity of adjusting trade-off between AbsBias@100 and Recall@100. For PBM, we introduce a trade-off parameter through a stochastic variable representing the probability of opting for a fair subset at any given time, as opposed to merely selecting the image with the highest similarity score. Additional details about this experiment are provided in the SM. Our results indicate that in a range of relatively high

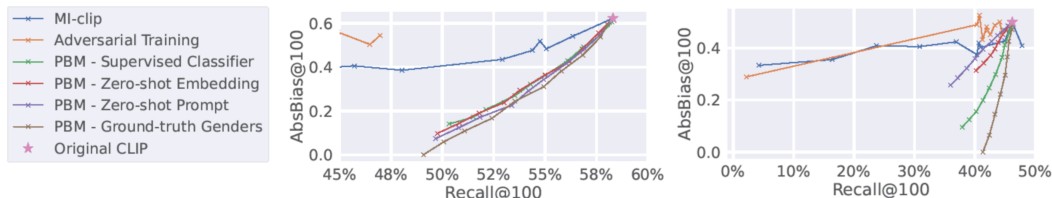

Figure 4: Trade-off between Recall@K and AbsBias@K for debiasing gender attributes within Occupation 1 (Middle) and race attributes using Occupation 2 (Right).

AbsBias@$K$ values, MI-*clip* and adversarial training are still able to reduce bias while preserving Recall@$K$. However, in terms of lowering AbsBias@$K$, they struggle to maintain satisfactory TBIR performance. In contrast, the proposed PBM consistently succeeds in sufficiently reducing bias and maintaining performance, provided that predictions of attributes are reasonably accurate. Details of implementing this trade-off can be found in the SM.

## 5 Discussion

In this study, we examined gender and racial bias in text-based image retrieval (TBIR) for neutral text queries. In an attempt to identify bias in the test-time (inference) phase, we conducted an in-depth quantitative analysis on bias reduction, alongside existing debiasing methods and the proposed PBM. We concluded that solely addressing training-time model-encoded bias is not sufficient for obtaining equal representation results, because test-time bias also exists due to imbalance in the test image set used during retrieval. So motivated, we proposed Post-hoc Bias Mitigation (PBM), a straightforward post-processing method that aims to directly alleviate test-time bias. Experiments on multiple datasets show that our method can significantly reduce bias while maintaining satisfactory retrieval accuracy at the same time.

Moreover, the potential impact of PBM extends far beyond the initial scope of text-based image retrieval systems. The core concept of our methodology can be seamlessly adapted to a wide variety of information retrieval systems, such as image-based text retrieval or query-by-example image retrieval, as long as the demographic information of the test set is accessible or can be estimated, *e.g.*, via zero-shot inference. Overall, our approach is not limited to enhancing fairness in text-based image retrieval, thus can be extended to a broad range of VL model applications.

## Limitations

Some limitations of the proposed method are duly acknowledged. Firstly, the efficacy of our approach is dependent upon the availability of a sufficient number of examples for each category (gender or racial attribute) within the test image set. Our work currently does not consider any techniques, such as using synthetic samples, to mitigate the issues arising from insufficient representations of certain demographic groups. Secondly, the debiasing effect of PBM pertains to the predictability and accessibility of demographic attributes. Attempts at debiasing religious representation, or other socio-cultural factors or identities, in images or speech present significant challenges, because the predicting or securing annotations regarding religious information, can be exceptionally difficult. Thirdly, as we prioritize the "equal representation", our retrieval results sacrifice recall performance to ensure a retrieval bag that contains equal representations of each demographic group. This compromises fairness towards content providers in the image retrieval process. From the standpoint of content providers, fairness should imply that similar samples are treated similarly, regardless of the demographic group membership of their provided samples. Lastly, our work assumes all queries are neutral. We do not develop a technique to identify if a query is neutral or not, thereby limiting the applicability of our method in hybrid text query retrieval where text query can be biased like in "Male doctor". It is important to note, however, that the above challenges are not unique to our method but are a common issue encountered in other debiasing approaches. A more comprehensive discussion on the limitations of our work is available in the SM.

## Acknowledgements

The authors would like to thank the anonymous reviewers for their insightful comments. This research was supported by ONR N00014-18-1-2871-P00002-3.

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

# Appendix

## A    Impact of Demographic Group Classifier on Debiased Results

Table 3: Group sensitivities and sensitivity ratios ($\rho$) for demographic attributes predicted by different classifiers on Occupation 1 - Gender and Occupation 2 - Race.

| Method | Gender | | | Race | | |
|---|---|---|---|---|---|---|
| | Male Sensitivity | Female Sensitivity | $\rho$ | Light skin Sensitivity | Dark skin Sensitivity | $\rho$ |
| PBM - Supervised Learning | 0.97 | 0.88 | 1.10 | **0.93** | **0.84** | 1.11 |
| PBM - Word Embedding | **0.98** | 0.94 | 1.04 | 0.84 | 0.78 | **1.08** |
| PBM - Zero-shot Prompt | **0.98** | **0.97** | **1.01** | 0.88 | 0.81 | 1.09 |

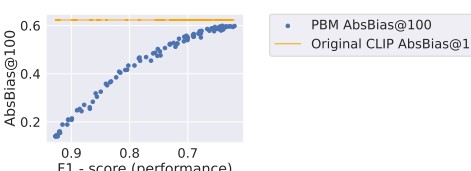
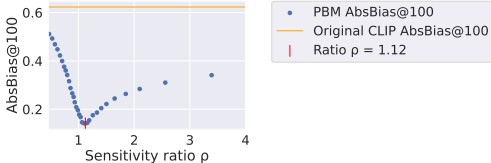

Figure 5: Relationship between the performance of the demographic group classifier (F1-score) and the retrieval bias (AbsBias@100) when utilizing PBM.

Figure 6: Relationship between the bias of the demographic group classifier (ratio of male sensitivity to female sensitivity) and the retrieval bias (AbsBias@100) when utilizing PBM.

The demographic group classifier is an important module of our proposed method PBM. The debiasing result is intricately linked to the demographic group classifier's accuracy and prediction bias towards different demographic groups. Figure 5 showcases the relationship between the demographic group classifier's performance and the ensuing retrieval bias, by artificially introducing noise to the demographic group (logit) predictions via Gaussian noise with a standard deviation ranging from 0 (no noise) to 1. These results underscore that better group classifier performance yields lower bias, that bias converges to that of the original CLIP as the group classifier gets worse, and importantly, that the bias after PBM will be no worse than that of the original CLIP.

Further, Table 3 shows the individual demographic group sensitivities under three different scenarios, from which we can see that the group classifier is i) able to achieve good classification sensitivity (no lower than 0.81 and 0.90 in average), likely because demographic image attributes (gender and skin tone) are typically captured in images, and ii) that different scenarios exhibit different degrees of bias as measured by the group sensitivity ratio, which must be close to 1 for the model to be unbiased. Table 3 reveals sensitivity discrepancies among different attributes. To delve deeper into the influence of classifier bias on PBM outcomes, we present the retrieval bias as a function of the sensitivity ratio in Figure 6. This is achieved by altering the gender classification threshold from 0 (maximizing male sensitivity) to 1 (minimizing male sensitivity). From Figure 6, we can conclude that the classifier bias does affect retrieval bias, however, only severely for more extreme sensitivity ratios, which is fortunately not the case in our results as shown in Table 3.

## B    Bias-recall Trade-off Strategies

In Figure 4, we exhibit the bias-recall trade-off curves for MI-clip, Adversarial Training, and various PBM methods. Here, we outline the missing details to achieve these trade-offs.

For adversarial learning, the trade-off is controlled by adjusting the adversarial loss weights between 0 and 1.0. In MI-clip, we modify the clipped dimensions from 10 to 500 (CLIP output dimension is 512). Regarding PBM methods, a trade-off parameter is introduced via a stochastic variable $\theta$, which denotes the likelihood of choosing a fair subset at any given time, instead of simply opting for the image with the top similarity score. Each curve is plotted by interpolating 10 points of the corresponding trade-off parameters.

## C   Datasets

**Occupation 1 (Kay et al., 2015)**   Occupation 1 comprises the top 100 Google Image Search results for 45 gender-neutral occupation terms, such as "chef", "librarian", "primary school teacher", *etc*. Each image within this dataset is annotated with a crowd-sourced gender attribute (either Male or Female) that characterizes the person depicted in the image. The entire Occupation 1 dataset is exclusively utilized for evaluating gender debiasing effect, as shown in Table 1.

**Occupation 2 (Celis and Keswani, 2020)**   Occupation 2 includes the top 100 Google Image Search results for 96 occupations. Each image in the dataset comes annotated with a gender attribute and a race attribute (represented by skin-tone, namely, Fair Skin and Dark Skin). Notably, the gender attribute and race attribute also include a N/A category, where the annotators have chosen the option of "Not applicable" or "Cannot determine" for the gender or skin-tone depicted in the image. Consequently, we treat the image labeled with N/A as a neutral example that does not contribute to the bias of retrieval, since the user cannot perceive gender or racial information from the image. Different from Occupation 1, gender attributes in Occupation 2 are categorized as {Male, Female, N/A}, while race attributes are classified as {Fair Skin, Dark Skin, N/A}. The enitre Occupation 2 dataset is only used for evaluation on mitigating gender and race bias, as the results shown in Table 1.

**MS-COCO (Lin et al., 2014)**   The first large-scale image-text dataset is MS-COCO captions dataset, which is partitioned into 113,287 training images, 5,000 validation images, and 5,000 test images. Each image is accompanied by five corresponding captions. Our experimental setup aligns with the methodology detailed by Wang et al. (2021a). Only the first caption of each image is used for evaluation. Further, they ensure all captions are gender-neutral by identifying and replacing or removing gender-specific words with corresponding neutral terms, with the help of predefined word banks (Zhao et al., 2017; Hendricks et al., 2018).

**Flickr30k (Plummer et al., 2015)**   The second large-scale image-text dataset employed in our experiment is Flickr30K, which contains 31,000 images obtained from Flickr. Adhering to the partitioning scheme presented in Plummer et al. (2015), we allocate 1,000 images each for validation and testing, with the remaining images designated for training. We obtain the ground truth of gender attributes of images in Flickr30k in the same way as MS-COCO (Wang et al., 2021a), as we detect the gender-specific words in the caption to determine the gender attributes of its corresponding image.

## D   Baseline Models

**Random Select**   To simulate an ideal scenario, where image features bear no dependency to gender (and race) attributes, for a neutral query $c$, we randomly select $K$ candidates from the true relevant image set $V_c^*$, with replacement . As for each query $c$, the size of the relevant image set $|V_c^*|$ is at most 100. Using sampling with replacement simulate the situation that the gender attribute distribution is fixed, and irrelevant to retrieval algorithm. We report AbsBias@$K$ for reference. The Recall@$K$ is omitted since the value is meaningless, as we only sample from the true relevant image set $V_c^*$.

**CLIP (Radford et al., 2021)**   We consider OpenAI's CLIP ViT-B/16 (Radford et al., 2021) as the VL model for all debiasing methods. Specifically, the image encoder $f_\phi(\cdot)$ is a Vision Transformer (ViT) (Dosovitskiy et al., 2020) comprising 12 transformer blocks of width 768, with 12 self-attention heads in each block. ViT processes images of size $224 \times 224$ by dividing them into $16 \times 16$ patches and outputs 512-dimensional image features by linear projection. The text encoder $f_\psi(\cdot)$ is a standard text transformer (Vaswani et al., 2017) with masked self-attention, consisting of 12 transformer blocks of width 512 and 8 self-attention heads in each block, with a linear projection layer at the end as well. The CLIP ViT-B/16 model is loaded with pre-trained weights provided by OpenAI (Radford et al., 2021). All the following debiasing methods use this pre-trained CLIP ViT-B/16.

**MI-*clip* (Wang et al., 2021a)**   MI-*clip* (Wang et al., 2021a) clips the fixed number of output dimensions of the image encoder in CLIP to reduce the mutual information between image features and demographic attribute distribution. For MI-*clip* in Table 1, we clip 312 dimensions of output image features. These 312 dimensions were chosen by examining the reduction in bias and while maintaining retrieval performance. We also show a trade-off between bias reduction and retrieval

performance of MI-*clip* with the number of clipped dimensions from 100 to 500 on the Occupation 1 dataset in Figure 4.

**Adversarial Training (Edwards and Storkey, 2015)** For adversarial training, we use the same minimax problem setup as in (Edwards and Storkey, 2015). The encoder is the original CLIP image encoder. The decoder is realized by a ViT with 8 vision transformer blocks, and the adversarial predictor is a 3-layer MLP. Training employs the same loss function in Edwards and Storkey (2015), where the final loss is the sum of the cost of reconstructing $v$ from $f_\phi(v)$, a measure of dependence between $f_\phi(v)$ and $g(v)$ and the error of target task (*i.e.,* image-text aligning loss $\mathcal{L}_{\text{NT-Xent}}$ (Radford et al., 2021)). We assign different weights to the loss of dependence measuring loss, in order to demonstrate the trade-off between bias reduction and retrieval performance. We report the adversarial learning results when the weight of measuring dependence loss is 0.7, and the weights for the other two losses are both 0.15. Additional weight combinations were considered and shown in Figure 4.

**Debias Prompt (Berg et al., 2022)** The Debias Prompt method (Berg et al., 2022) also leverages an adversarial learning framework. However, instead of just fine-tuning the image and text encoders, they also prepended zero-initialized learnable prompts before inputting query tokens. Considering that their debias-prompt model is already debiased for gender and race attributes, we directly evaluate their pre-trained model (sourced from their github repository `https://github.com/oxai/debias-vision-lang`) on the Occupation 1 and Occupation 2 datasets.

**CLIP-FairExpec (Mehrotra and Celis, 2021)** We tailor FairExpec (*not originally proposed* for TBIR) to our task by integrating it with CLIP and our proposed gender predictor $\hat{g}(\cdot)$. We refer to this adaptation as CLIP-FairExpec in our experiments.

Using binary gender as an example for simplicity, our CLIP-FairExpec treats the image-text similarity output as the utility score for each image. Then, the objective of the optimization is to maximize the total similarity scores for selecting $K$ images corresponding to a query $c$. The noise estimate $q$ in the original FairExpec is derived from the probability output of our attribute predictor $\hat{g}(v)$. We use the probability output from the attribute predictor $\hat{g}(v)$ as the noise estimate $q$. Also, there is a constraint on the sum of the noise estimates $q$ such that the sum is at least $L - \delta K$ and at most $U - \delta K$, where $L$ and $U$ is the lower bound and upper bound for the sum of the noise estimate, respectively. $\delta \in (0, 1)$ is a noise tolerance level, that controls how much the constraints can be violated due to the presence of noise. Since our fairness objective is equal representation for each gender attribute class, we wish the sum of noise estimate for each class of gender attribute is equal. Hence, we set the $L = U = K/2$. In order to force our model to prioritize minimizing bias over maintaining performance, we choose a very small $\delta = 0.001$. We select $K$ images from $\mathcal{V}$ with respect to a neutral query $c$ based on the above constrained optimization problem. Further, each selection is solved by the GUROBI solver (Gurobi Optimization, LLC, 2023). Upon solving for all selections for queries in $\mathcal{C}$, we compute the AbsBias@$K$ and Recall@$K$ presented in Table 1.

**SCAN (Lee et al., 2018)** We consider the Stacked Cross Attention Network (SCAN) (Lee et al., 2018) as an alternative VL model to CLIP. SCAN is a specialized in-domain training model, so it is trained on the MS-COCO training dataset and tested on the MS-COCO test dataset. Similarly, for the experiments with Flickr30k, the model is trained on the Flickr30k training set and then tested on the Flickr30k test set. We use official implementation of SCAN from `https://github.com/kuanghuei/SCAN`.

**FairSample (Wang et al., 2021a)** To mitigate bias during the training of SCAN, we implement the FairSample approach as recommended by Wang et al. (2021a). We maintain the same hyperparameters settings as Lee et al. (2018). To address the bias arising from the unbalanced gender distribution within training batches, FairSample is proposed in the following way: for every positive image-text pair $(v, c)$ within a training batch, we first identify if the query $c$ is gender-neutral or gender-specific. If the training query $c$ is gender-neutral, a negative image is sampled from either the male or female image sets, each with a probability of 1/2. However, if the query is gender-specific, we maintain the original negative sampling strategy, thereby preserving the model's ability to generalize effectively on such queries.

# E  Post-hoc Bias Mitigation (PBM)

## E.1  Engineering Details

**PBM - Supervised Classifier**  We can determine the gender attributes with a pre-trained image classifier. Here, the image classifier is pre-trained on MS-COCO training set with gender attribute annotations from Zhao et al. (2021). The image classifier is a 3-layer multi-layer perceptron (MLP) as shown in Table 5, that takes the image representation from the original CLIP as input. We empirically show that the image classifier can be highly accurate even using a light-weight classification MLP. The F1-score for gender attribute prediction is 92.8%.

**PBM - Zero-Shot Embedding**  We describe the first of the two types of zero-shot inference described in Section 3.5. For zero-shot inference based on the embedding approach, we choose the text embeddings for {"Unknown Gender", "Man", "Woman"} tokens to classify the gender attributes of images, and {"Unknown Skin", "Fair Skin", "Dark Skin"} for categorizing race attributes. The gender or race attribute of an image $v$ is determined by which text embedding has the maximum similarity score to the image representation $f_\phi(v)$.

**PBM - Zero-Shot Prompt**  For the second zero-shot inference described in Section 3.5, namely, the prompt method, we prepend adjectives to the text query $c$. We use {"", "Male", "Female"} for gender attributes and {"", "Fair-skinned", "Dark-skinned"} for race attributes. The gender attributes for each image retrieved by the query $c$ is determined by which prompted query has the maximum similarity score to the image representation $f_\phi(v)$.

**PBM - Ground-Truth Attribute (Gender or Skin-tone)**  We use the annotations in the dataset as the predicted attributes $\hat{g}(v)$ for reference. This shows the upper-bound performance of our method if all gender predictions are correct (known).

## E.2  Additional PBM Results

In Table 4, we showcase the results of applying PBM to CLIp models that has been debiased by other approaches, such as MI-clip, Adversarial Learning, and Debias Prompt. When PBM is utilized in conjunction with other debiasing strategies, it exhibits a unique bias-recall trade-off, thus catering to a variety of application scenarios.

Table 4: Results of applying PBM - Supervised Learning on modified or fine-tuned CLIP.

| Method | Occupation 1 - Gender | | Occupation 2 - Race | |
|---|---|---|---|---|
| | AbsBias@100 ($\downarrow$) | Recall@100($\uparrow$) | AbsBias@100($\downarrow$) | Recall@100($\uparrow$) |
| **PBM** | .1404 | 50.3 | .0955 | 37.9 |
| MI-*clip* - **PBM** | **.0780** | 42.1 | **.0737** | 29.1 |
| Adversarial Training - **PBM** | .1000 | 39.6 | .0997 | 35.7 |
| Debias Prompt - **PBM** | .1711 | **52.1** | .1035 | **40.6** |

# F  Neural Network Architectures

We summarize the details of the neural networks employed in our experiments in Table 5. For the Image Encoder, the Patch Extraction (dimensions: 16,16) extracts 196 non-overlapping $16 \times 16$ patches from the 224×224 image. These extracted patches are subsequently flattened. The subsequent Positional and Linear Embedding (768) maps these patch vectors onto a 768-dimensional space and adds 2D positional embeddings of patches to the 768-dimensional vectors. Next, 12 Vision Transformer Blocks (768, 12) processes the 768-dimensional embeddings. Each of these blocks features 12 self-attention heads. Lastly, the output embedding is obtained from a unique classification token ([CLS]) that we add to the input sequence of patch embeddings. The output from [CLS] Token $1 \times 768$ is then reduced from 768 dimensions to 512 dimensions using a Linear Projection (512).

Similarly in the Text Encoder, the initial phase involves Positional and Token Embedding (512). This step maps each token in the input text onto a 512-dimensional vector space and integrates positional embeddings into these vectors. Following this, the text encoder employs 12 Transformer Blocks

Table 5: The architecture of each component of CLIP and the MLP used in our experiments.

ImageEncoder(·)

| Layer | Type |
|---|---|
| 1 | Patch Extraction(16, 16) |
| 2 | Positional and Linear Embedding(768) |
| 4 - 15 | Vision Transformer Blocks(768, 12) |
| 16 | [CLS] Token $1 \times 768$ |
| 17 | Linear Projection (512) |

TextEncoder(·)

| Layer | Type |
|---|---|
| 1 | Positional and Token Embedding (512) |
| 2 - 13 | Transformer Blocks (512, 8) |
| 14 | [CLS] Token $1 \times 512$ |
| 15 | Linear Projection (512) |

MLP(·)

| Layer | Type |
|---|---|
| 1 | fc-512 + BatchNorm + ReLU() |
| 2 | fc-512 + BatchNorm + ReLU() |
| 3 | fc-512 + BatchNorm + ReLU() |
| 4 | fc-n_class + Softmax() |

(512, 8) to process these 512-dimensional embeddings. Each of these blocks contains 8 self-attention heads. Finally, the output embedding is derived from [CLS] Token $1 \times 512$. The subsequent Linear Projection (512) then maps the extracted text representation onto the multi-modal embedding space that aligns with the image embeddings.

## G   Computation Resources

All of our experiments ran on one NVIDIA TITAN Xp 12GB GPU with CUDA version 11.5.

## H   Code and Data Availability

Occupation 1 dataset is available at `https://github.com/mjskay/gender-in-image-search`.

Occupation 2 dataset can be downloaded from `https://drive.google.com/drive/folders/1j9I5ESc-7NRCZ-zSDOC6LHjeNp42RjkJ`.

MS-COCO dataset can be access through `https://cocodataset.org/#home`, and its crowd-sourced gender/racial annotations from `https://princetonvisualai.github.io/imagecaptioning-bias/`.

Flickr30k dataset can be access via `https://shannon.cs.illinois.edu/DenotationGraph/`. And the gender word banks to identify the gender attributes of Flickr30k's images is avaliable in the Appendix of the paper by Wang et al. (2021a).

## I   Broader Impact

The recent years constituting what can be called the model architecture unifying era, witnessed a seismic shift from small task-specific models to foundation models containing billions of parameters, with numerous applications deployed based on such large models. However, as artificial intelligence (AI) systems become more prevalent, the challenging question of fairness becomes more urgent. The concept of fairness in machine learning revolves around creating algorithms and models that DO NOT discriminate against certain groups based on gender, race, socioeconomic status, or any other potentially biasing factors. As machine learning algorithms are increasingly used in decision-making

processes, from job applications, college admissions, to criminal justice and healthcare, subsets of the population who represent minorities may see unfavoring model performance compared to individuals in majority groups. Therefore it is imperative to develop unbiased machine learning systems such that decisions are made fairly and equitably. Our study concerning fair image retrieval, among many other fairness research works, can be used to inform policymakers about the potential risks and benefits of AI systems, potentially enacting new laws and regulations to ensure that these systems are utilized responsibly and ethically.

Specifically, the biased performance of a model is possibly caused by statistical skewness both in the training and testing sets. Existing methods mainly focus on enforcing independence between the model's output and sensitive attributes during training. However, much less effort has been made to mitigate bias during test-time, a potentially vital component of the debiasing procedure. Many machine learning systems are deployed in a setting where the biased testing set is almost guaranteed, and as such, may suffer from fairness concerns. Importantly, PBM is able to dissociate the ranking similarity from sensitive/protected attributes (*e.g.*, gender) thus reducing bias, meaning that image candidates share an equal chance to be retrieved even in an unbalanced testing set. We do not claim that PBM guarantees fairness, and there is always the risk that it may be misinterpreted or exploited, but we hope that PBM encourages a more inclusive approach to AI development.

