# OpenReview forum: "Mitigating Test-Time Bias for Fair Image Retrieval"
_NeurIPS.cc/2023/Conference — NeurIPS 2023 poster_

### Official Review · Reviewer_PMeK · 2023-06-29

**Soundness:** 3 good
**Presentation:** 3 good
**Contribution:** 2 fair
**Rating:** 6
**Confidence:** 4

**Summary:**

The paper considers fair text-based image retrieval, in particular when the target retrieval set is biased. The chosen fairness metric is _equal representation_, e.g., the number of retrieved male and female images should be identical. The authors conjecture that if the target retrieval set is biased, debiasing the (multimodal vision-language) model during training is not sufficient to guarantee fairness of the retrieved results in test-time. Therefore, they propose Post-hoc Bias Mitigation (PBM): first, they sort the most relevant candidate images from each group (e.g., male and female) separately and then take equal amount of images from the two groups. In their experiments they consider different ways of estimating the (unknown at test time) sensitive attributes: zero-shot embedding, prompting and supervised classification. Experiments are executed on 4 standard datasets: Occupation 1 and 2, MS-COCO and Flickr30k.

**Strengths:**

* The paper is well-written and easy to follow.
* The prior work is presented fairly and it is somewhat clear how this paper is positioned with respect to it. It touches on a relevant topic with potentially high social impact.
* The presented solution to the proposed problem formulation is intuitive, straightforward and seems to work well in practice.

**Weaknesses:**

1. The idea of post-processing is not novel and the technical contribution of the proposed solution is limited and rather incremental, in my opinion. Hardt et al. (2016) and Calders and Verwer (2010) propose probability thresholding, while Karako and Manggala (2018) consider a similar procedure of iterative selection of retrieved images based on their relevance and diversity. I acknowledge that the problem settings in these works are sufficiently different from the current paper, but the idea that we can take an equal amount of images from the two demographic groups (sorted by relevance) if we want to equalize the number of images from both groups, is not particularly exciting or insightful, even if the demographic attributes are inferred in a zero-shot manner during test-time.

Calders and Verwer, Three naive Bayes approaches for discrimination-free classification, 2010\
Karako and Manggala, Using Image Fairness Representations in Diversity-Based Re-ranking for Recommendations, 2018

2. It seems to me that the key factors for the performance (in terms of fairness; for a given, fixed query) of the method are the distribution of male/female among the ground-truth samples in the target retrieval set as well as the accuracy of $\hat{g}(\cdot)$ - the sensitive attribute predictor. I think it may be informative to further explore what happens when the distribution of samples from each demographic group is varied or when $\hat{g}(\cdot)$ is not very accurate. For example, according to Ghosh et al. (2021), "developers should not use inferred demographic data as input to fair ranking algorithms, unless the inferences are extremely accurate".

Ghosh et al., When Fair Ranking Meets Uncertain Inference, SIGIR 2021

3. The details how to control the fairness/utility trade-off seem to be missing from the supplementary material (L350). I am unable to find other details from the SM as well (e.g., adversarial learning results, L187).

**Questions:**

Q1: I think it will be prudent to show the distribution of the sensitive attributes for the target ground-truth retrieval set(s) ($V_{c}^{\*}$) as well as the whole test sets ($\mathcal{V}$), e.g., the male:female ratio per occupation in the Occupation datasets. I would be interested to see the relationship between bias/recall and the distribution ratio (between male/female in $V_{c}^{\*}$). For example, if the target retrieval set is relatively balanced, are the other baseline methods still biased?

Q2: How do you control the bias-recall trade-off? Do you vary the probability (L348) from 0.1 to 0.9 on Figure 4? \
Minor: Are the results in Figure 4. for Occupation 1 and 2 datasets and gender as the sensitive attribute?

Q3: How would you combine multiple sensitive attributes (e.g., if we want to be fair w.r.t. both gender and race in Occupation 2)?

Q4: Is the problem formulation and the metrics standard/adapted from prior work or novel/custom? I see that you closely follow (Wang et al., 2021a), so might be good to mention this.

Q5: Formal metric definitions. For Recall@K why don't you consider $\frac{1}{|C|}\sum_{c \in C} \frac{| V_{c,K} \cap V_{c}^* |}{|V_{c}^{*}|}$ ? Or it does not matter as $ | V_{c}^{\*} | = K$ is the same for all queries $c$? When computing the bias, why do you take the absolute value, instead of considering Bias@K, following (Wang et al., 2021a)? If you take the absolute values, how do you get negative gender bias in Figs. 2 and 3?

Q6: When instantiating PBM, do you use the original CLIP as image/text encoders, or any of the other baselines MI-clip, Adversarial training, Debias Prompt? What is the effect if you instantiate PBM with any of these "debiased" VL models?

Q7: What is the underlying reason for the recall to still be low? Are the VL models not good enough? What happens if we lower K and consider the precision, esp. for the Occupation datasets? Unless male:female ratio is 50:50, it is impossible to obtain 100% recall for the given query anyway, as we have to retrieve other images from the underrepresented group that do not correspond to the occupation for which we query. L320: should it be 61:39?

**Limitations:**

In general, I suggest trying to (partially) include the Limitations section in the main paper or at least refer to it.

Lim1: Fairness metric definition. Translated in the context of supervised classification, it would look like $p(g = 0 | \hat{y} = 1) = p(g = 1 | \hat{y} = 1)$, where the value of the sensitive attribute $g$ is unknown at test time and "prediction" of $\hat{y} = 1$ corresponds to the image being retrieved. This does not look like any other "standard" (supervised classification) fairness metric. I appreciate that the authors have included motivation about this design choice, but maybe some further discussion is needed. I would like to point out that in some cases or other scenarios the chosen metric may lead to unfairness, e.g., for the content providers (if a sample with lower utility is preferred over another solely due to having the "correct"/"desirable" sensitive attribute).

Lim (comment) 2 [out of scope]: Depending on the setting, if there are insufficient number of samples with a particular sensitive attribute, synthetic samples might be generated (Tanjim et al., 2022).

Tanjim et al., Generating and Controlling Diversity in Image Search, WACV 2022

Lim (comment) 3 [out of scope]: An end-to-end pipeline would need to decide if the input query is gender-neutral or not. E.g., querying for a famous actor/actress is clearly not gender-neutral. Then, is gender neutrality a 0/1 binary property, or it can be scored?

---

> ### Author Rebuttal · Authors · 2023-08-09
>
> ### **Response to Reviewer PMeK**
> Thanks for your detailed review and constructive feedback. Before reading our dedicated response to your comments, we kindly suggest you start with our global response, which offers a broad overview and addresses some of the general concerns raised.
> Below is our response to your comments,
>
> #### **"The idea of post-processing is not novel and the technical contribution of the proposed solution is limited..."**
> While we acknowledge that using post-processing method to reduce the output bias is not novel, our work contributes to leveraging a group classifier built by supervised learning, zero-shot word embedding and zero-shot prompts. Our approach effectively reduces the bias in retrieval, showcasing the utility of neural network predictions for gender and race attributes in addressing retrieval test-time bias. Calders and Verwer (2010) requires ground-truth gender attribute and Karako and Manggala (2018) obtain the gender labels from user feedback data. We will clarify more about our contribution and add these two papers in the related work of our revision.
>
> ####  **"It seems to me that the key factors for the performance (in terms of fairness; for a given, fixed query) ....".**
> The distribution of the retrieval pool among Occupation 1 & 2, COCO and Flikcr30k are different. We list the male rates for these datasets in Table 4 (rebuttal).
>
> Though Ghosh et al. (2021) states "developers should not use inferred demographic data as input to fair ranking algorithms, unless the inferences are extremely accurate", our work focuses on mitigating perceivable bias in web image search system. The performance of our demographic group classifier is less likely to be poor given that perceivable biases are in principle easy to classify. This is demonstrated in Table 1 (rebuttal). Moreover, our work argues that only the post-hoc group classifier can address the test-time bias in image retrieval, while none of other debiasing methods addressed the test-time bias scenario.
>
> #### **"The details how to control the fairness/utility trade-off..."**
> Thanks for pointing it out. We will include these details in the revision. To control the trade-off of adversarial learning, we vary the weights of adversarial loss from 0 to 1.0. For MI-clip, we vary the clipped dimensions from 10 to 500 (CLIP output dimension is 512). We include these details in the Section C.2 of our Supplemental Material. We will include a dedicated section to describe trade-off strategies of PBM, MI-clip and adversarial learning, including pseudo-codes of trade-off strategies and how to choose $\theta$ under different application context, to avoid confusions.
>
> #### **"Q1: I think it will be prudent to show the distribution of the sensitive attributes "**
> We show the distribution of gender across Occupation 1 & 2, COCO-val and Flickr30k test in Table 4 (rebuttal). If the target retrieval set is balanced, we believe there is likely still bias in the retrieval set since sampling bias exists if the retrieval bag size is not large enough.
>
> #### **"Q2: How do you control the bias-recall trade-off? "**
> To manage the bias-recall trade-off, we employ a parameter θ, that is the probability of choosing a fair subset at any given time, rather than solely selecting the image with the highest similarity score. We vary θ in increments of 0.1, ranging from 0.0 to 1.0. As illustrated in Figure 4, the left-side plot is for Occupation 2 - Race, while the right-side plot for Occupation 1 - Gender.
>
> #### **"Q3: How would you combine multiple sensitive attributes ... "**
> In this case, we can define combinations of different attributes from which PBM can choose separately. For example, we can define the combination each time PBM has to choose: {"Light skin male", "Dark skin female"} or {"Light skin female", Dark skin male} or {"Light skin male", "Dark skin female", "Light skin female", Dark skin male}. Extending to combine multiple sensitive attributes is an intriguing direction which we consider for future work.
>
> #### **"Q4: Is the problem formulation and the metrics standard/adapted from prior work or novel/custom?"**
> Indeed, our work is inspired by (Wang et al., 2021a). In Table 2(paper), we use the same metric and setup as (Wang et al., 2021a). We do not use Bias@K in our main result(Table 1 of paper) because we contend that the Bias@K used in (Wang et al., 2021a) omits using the absolute value in the sum. The Bias@K metric can be misleading sometimes because the positive bias and negative bias cancel each other out in the summation. This will be emphasized in the Section 4 of revision.
>
> #### **"Q5: Formal metric definitions. "**
> For Recall@K, we follow the Recall@K used in (Wang et al., 2021a). $\frac{1}{|C|}\sum_{c \in C} \frac{| V_{c,K} \cap V_{c}^* |}{|V_{c}^{*}|} $ is also a valid metric but should be named as "mean Recall@K". We choose to use AbsBias@K in Table 1 (paper) because we argue that the Bias@K used in (Wang et al., 2021a) omits using the absolute value in the sum.
>
> In Figure 2 and 3 (paper), the gender bias is a simple summation of gender attributes (male = +1 and female = -1) in different quantiles. We will clarify these points in the Section 3.3 of revision.
>
> #### **“Q6: When instantiating PBM, do you use the original CLIP as ...”**
> In our reported results, we apply PBM on the original CLIP. We also report applying PBM on other fine-tuned models in Table 2 (rebuttal).
>
> #### **"Q7: What is the underlying reason for the recall to still be low?"**
> We conduct a brief analysis of the low recall performance in COCO and Flickr30k in the global response.
>
> We will acknowledge this point in the limitation section of our revision.

---

> > ### Comment · Reviewer_PMeK · 2023-08-15
> >
> > I thank the authors for their clarifications and for providing further data and details. I still believe that the technical novelty is limited (and more rigorous datasets and setups are required to drive this problem further) but the major questions raised by me and most of the reviewers were addressed during the rebuttal, so I increase my score (to 5).
> >
> > I still hope that the authors can answer the following question (Q1): if the retrieval set is balanced, do the existing methods still result in more biased retrievals than PBM?
> >
> > Moreover, I also hope that the authors will acknowledge some of the limitations brought by me. In particular, (Limitation 1) some fairness decisions can still lead to unfairness (e.g., preferring a sample with lower utility over another solely due to having the "correct"/"desirable" sensitive attribute might be unfair and discriminatory to the content providers [1]). I find this particularly relevant to the discussion of the bias/recall tradeoff and the lower recall results on COCO and Flickr30k.
> >
> > [1] Wang et al., A survey on the fairness of recommender systems, ACM Transactions on Information Systems, 2023

---

> > > ### Author Response · Authors · 2023-08-17
> > >
> > > Thank you for acknowledging our efforts in our rebuttal. We acknolwedge the limitations(1,2,3) highlighted in your comments and include them into our limitation section. Below is our response to your follow-up questions:
> > >
> > > #### *"(Q1): if the retrieval set is balanced, do the existing methods still result in more biased retrievals than PBM?"*
> > >
> > > Ideally, when the retrieval set is balanced, the retrieval bias of existing in-processing methods should be no more than our PBM's result, if the retrieval bag size is sufficiently large. To justify this idea, we conduct an additional debiasing experiment on an artificially balanced subset of Occupation 1 and Occupation 2. Here, we focus on the binary gender scenerio for simplicity.
> > >
> > > To ensure that the retrieval set corresponding to each query is balanced, we select images related to occupations that have relatively balanced gender representation in the original dataset. These occupations include 'chemist', 'pharmacist', 'insurance sales agent', 'biologist', 'author', 'cook', 'real estate agent' and 'doctor' . The resulting subset comprises 322 male and 326 female. Below, we show the debias results using in-processing methods such as MI-clip, Adversarial Learning and Debias Prompt, comparing to our PBM - Supervised Classifier method:
> > >
> > > |                             | AbsBias@100 | Recall@100 |
> > > |-----------------------------|-------------|------------|
> > > | Original CLIP               | .0950       | 39.0       |
> > > | MI - Clip                   | .0550       | 34.9       |
> > > | Adversarial Learning        | .0675       | 33.8       |
> > > | Debias Prompt               | .1025       | 38.4       |
> > > | PBM - Supervised Classifier | *.0525*       | *39.0*       |
> > >
> > > From the results, we observe that the resulting bias of MI-clip and Adversarial Learning is very close to our PBM, thereby justifying the idea that if there is no dataset bias, in-processing methods should work as well as PBM. Debias Prompt does not perform well. We hypothesize that it is due to its debiasing effect failing to generalize to Occupation 1 and Occupation 2 dataset, consistent with the results shown in Table 1(paper).
> > >
> > > Interestingly, it is should be noted that in a balanced dataset, the PBM does not exhibit a drop in recall performance compared to the original CLIP. This is because the ground truth retrieval set aligns with the equal representation that PBM pursues. In contrast, MI-clip and Adversarial Learning experience performance drops due to their adverse effects on the feature space of the retrieval model. Thereby, when a balanced retrieval set is presented, employing PBM is also a promising choice, as it is less likely to have a negative impact on the retrieval performance.
> > >
> > > #### *"Moreover, I also hope that the authors will acknowledge some of the limitations ..."*
> > >
> > > Yes, we sincerely acknowledge the limitations (1, 2, 3) that you have highlighted in your comments.
> > >
> > > Specifically, for limitation 1, as we prioritize the "equal representation", our retrieval results sacrifice recall performance to ensure a retrieval bag that contains equal representations of each demographic group. From the standpoint of content providers, fairness should imply that similar samples are treated similarly, regardless of the demographic group membership of the sample. The former one(ours) is pertinant to group fairness[1], while the latter one(content providers' perspective) is more closely aligned with individual fairness[2]. The intrinsic conflict between group fairness and individual fairness[3] presents a significant challenge, making it difficult for our work to optimally address both. Determining the appropriate trade-offs between these two aspects of fairness is an important direction for future work.
> > >
> > > For limitation 2, we acknowledge that our work does not consider any techniques, such as using synthetic sampels, to mitigate the issues arising from insufficient representations of certain demographic groups. Addressing the bias in insufficient retrieval pools is a direction we plan to explore further.
> > >
> > > For limitation 3, we acknowledge that currently our work does not incorperate any techniques to identify if a query is neutral or not. Also, we acknowledge our assumption of gender neutrality being binary is limited. Whether the neutrality of a text query should be considered as binary or as continuous is an intriguing question to explore.
> > >
> > > Again, thank you for your insightful comments and suggestions. We are committed to continually refining our work.
> > >
> > > [1] Chouldechova, Alexandra, and Aaron Roth. "A snapshot of the frontiers of fairness in machine learning." Communications of the ACM 63.5 (2020): 82-89.
> > >
> > > [2] Dwork, Cynthia, et al. "Fairness through awareness." Proceedings of the 3rd innovations in theoretical computer science conference. 2012.
> > >
> > > [3] Binns, Reuben. "On the apparent conflict between individual and group fairness." Proceedings of the 2020 conference on fairness, accountability, and transparency. 2020.

---

> > > > ### Comment · Reviewer_PMeK · 2023-08-17
> > > >
> > > > Thank you. The extra details make me more confident in my score (increasing to 6). I hope the authors will be able to incorporate these clarifications as well as the suggestions from the discussions with the other reviewers in their final revision.

---

### Official Review · Reviewer_VHZT · 2023-07-02

**Soundness:** 3 good
**Presentation:** 3 good
**Contribution:** 3 good
**Rating:** 6
**Confidence:** 3

**Summary:**

The paper tackles the problem of bias in text-image retrieval. The goal is to generate fair and unbiased image retrieval results given neutral textual queries. As opposed to previous approaches that aim to separate the learned representations of images and text from characteristics such as gender and race, the paper claims that these methods are insufficient for achieving the objective of equal representation, primarily because bias often persists during the evaluation phase within the target retrieval set. In order to address this, the authors propose PBM, a straightforward and efficient test-time post-processing debiasing algorithm that generates fair retrieval subsets, guided by predicted gender/race information obtained from an off-the-shelf classifier or inferred via zero-shot using the VL model itself and evaluate their approach on various benchmarks.

**Strengths:**

The paper tackles an important problem. I think that addressing bias is a very important direction and this paper can provide useful insights to the research community.

**Weaknesses:**

My main concern with the paper is that I don't understand if there is a link between the performance of the gender/race classifier and the bias. Since the performance of these classifiers directly affect the performance, I would have wished to see more ablations on this front.

Some parts seem to be unpolished. For example Fig 4 is very hard to read, especially the last plot. Also, the colors are quite similar and it's very hard to assess which method is which.

**Questions:**

1. What architecture do you use for the text encoder and video encode? I think this information should be added in sec 3.2

2. What is the performance of the gender prediction? (both zero-shot and supervised) Does the zero-shot prompt works better than supervised classifier? The results from Tab 1 seem to me to be in contradiction with Fig 4 (if I am reading it correctly) since AbsBias is always lower for supervised vs zero-shot, but in Tab 1 the results are different.

3. Any insights on why the performance drop is so large in Tab 2?

Minor - Can the method be applied to other types of biases except gender and race biases? Are there any other datasets that have other bias information? What is needed to adapt the method to other biases?

**Limitations:**

The limitations are discussed and a brief societal impact is also presented.

---

> ### Author Rebuttal · Authors · 2023-08-09
>
> ### **Response to Reviewer VHZT**
> Thanks for your detailed review and constructive feedback. Before reading our dedicated response to your comments, we kindly suggest you start with our global response, which offers a broad overview and addresses some of the general concerns raised.
> Below is our response to your comments,
>
> #### **"...if there is a link between the performance of the gender/race classifier and the bias..."**
> Thank you for bringing up this critical point. We understand your concern regarding the relationship between the performance of the demographic group classifier and the overall bias of our system. In Table 1 (rebuttal), we show the performance (sensitivity) of the group classifier stratified across different demographic groups. The Zero-shot Prompt method achieves best sensitivities on gender, whereas Supervised Classifier is the best on race. Combining these results with the AbsBias@100 reported in the Table 1 (paper), we can conclude that the debiasing effect is consistent to the performance of the group classifier.
>
> Moreover, in Figure 1 and Figure 2 (rebuttal), we examine the relationship between the group classifier predictions and retrieval bias, by varying the performance and sensitivities of the group classifier. Figure 1 (rebuttal) demonstrate that better group classifier performance leads to reduced bias, that bias converges to that of the original CLIP as the group classifier gets worse, and importantly, that the bias after PBM will be no worse than that of the original CLIP. Meanwhile, Figure 2 reveals that the retrieval result will have more bias as the group classifier has more predictive disparity (i.e., sensitivity ratio approaches 0 or $\infty$), and the lowest bias is achieved when the group classifier prediction is fairly balanced (i.e., sensitivity ratio close to 1).
>
> Further, for a more theoretical justification, please refer to the response to reviewer TTMH.
>
> #### **"Some parts seem to be unpolished..."**
> Thanks for pointing out the visual quality of our paper. The performance of different PBM methods is very similar in Figure 4. Hence, it is somewhat difficult to distinguish them. We improve the readability of Figure 4 by zooming into the bias-recall curves for PBM variants. Please refer to Figure 3 (rebuttal). We will include the better visualization in our revision.
>
> #### **"What architecture do you use for the text encoder and video encode?”**
> The image encoder is a ViT with 8 vision transformer blocks and the text encoder has 12 Transformer Blocks. Both image and text encoder are pre-trained by OpenAI [1]. The details of the network architecture is included in our Supplementary Material. We will certainly include this information into Section 3.2.
>
> [1] Radford, Alec, et al. "Learning transferable visual models from natural language supervision." ICML (2021).
>
> #### **"What is the performance of the gender prediction?..."**
> We include the sensitivity across different demographic groups in the Table 1 (rebuttal). For the Figure 4 (paper), the left plot is the recall/bias trade-off evaluated on debiasing race for Occupation 2 dataset, which should correspond to the third column in Table 1 (paper). In this case, the supervised classifier predicts more accurately than the zero-shot prompt method, so the lowest AbsBias@100 is achieved by the PBM - Supervised Classifier in Occupation 2 - Race. This will be emphasized in the caption of Figure 4 in revision to avoid confusion.
>
> #### **"Any insights on why the performance drop is so large in Tab 2?"**
> For the COCO and Flickr30k benchmarks, the retrieval pools for each query are very limited, thus when we apply PBM, the recall performance suffers because there are usually not enough minor demographic groups to satisfy the PBM constraint.
>
> #### **"Minor - Can the method be applied to other types of biases except gender and race biases?"**
> Yes. We believe our method can be applied to other types of bias as long as such bias are perceivable, which means they can be annotated by a human and predicted by a neural network. UTKface and Fairface are face datasets that are annotated with age, gender and race [2,3]. Our methods can definitely be applied to debias age when retrieving images from UTKface. We will include experiments with UTKface and Fairface in our revision (we were unable to get the results in time before the rebuttal deadline).
>
> [2] Zhang, Zhifei, Yang Song, and Hairong Qi. "Age progression/regression by conditional adversarial autoencoder." CVPR (2017).
>
> [3] Karkkainen, Kimmo, and Jungseock Joo. "Fairface: Face attribute dataset for balanced race, gender, and age for bias measurement and mitigation."  CVPR (2021).

---

> > ### Comment · Reviewer_VHZT · 2023-08-17
> > **Rebuttal**
> >
> > I acknowledge that I have read the rebuttal and the author response.
> >
> > I appreciate the detailed response that the authors made. While I think there still are things that need to be changed, the authors promised to address these in the final version, hence I raise my score to Weak Accept.

---

> > > ### Author Response · Authors · 2023-08-17
> > >
> > > Thank you for your time and efforts in carefully reading our rebuttal and reconsidering the score of our paper. We are committed to making the necessary changes in the revision as promised.

---

### Official Review · Reviewer_TTMH · 2023-07-06

**Soundness:** 1 poor
**Presentation:** 2 fair
**Contribution:** 3 good
**Rating:** 6
**Confidence:** 4

**Summary:**

The paper proposes a method for mitigating test-time bias in image retrieval of vision-language models. More concretely, the authors demonstrate how prior work debiasing methods fail to ensure equal representation of different sensitive groups in the retrieval bag at test-time when the test set is imbalanced across different sensitive groups (for example, there are more male engineers at the test set than female engineers). To mitigate this issue, they propose a simple method where they train a sensitive group classifier (for example, a gender classifer) to split the test set across the sensitive groups and then perform the retrieval over each subset independently (if the task is to retrieve K samples, and the sensitive groups are: Male/Female then they would retrieve K/2 from the predicted Male group and K/2 from the predicted Female group). They show through experiments that their method reduces the disparity between the number of retrieved Male and Female samples at test time while maintaining “satisfactory” retrieval performance.


**Strengths:**

1- The problem is well motivated; indeed, when the test set is imbalanced, it is clear that the prior work debiasing method, while not amplifying the bias of the test set, will nevertheless exhibit the proportional bias of the test set.

2- The method is simple and easy to implement.

3- The writing is clear, and the results are well explained.


**Weaknesses:**

This work's main weakness is using a sensitive group classifier (for example, gender classifier) to split the test set into different subgroups. An apparent problem with this approach is that the sensitive group classifier will likely be biased. This is because, similar to how VL models correlate class (for example, professions) with gender, it will also correlate gender with class. So, for example, if the dataset has more Male engineers, then a gender classifier based on the VL model features is more likely to correctly predict engineers as male than predicting engineers as Females. This apparent problem is not discussed in the paper not are solutions proposed to address it.

Furthermore, the experimental results could explain why this might be an issue. More concretely, the proposed method consistently reduces the recall performance, often by significant margins. For example, in Table 2, Recall 1 is reduced from 67.2 to 41.2 on Flickr30k. This pattern might be directly related to the bias of the gender classifier. Indeed, if the classifier, for example, is not good at predicting Female Engineers as Female, then the test set predicted Female split might contain very few engineers, and thus, the K/2 extracted samples from the predicted split might contain very few relevant engineers and thus reducing the recall performance.


**Questions:**

Given the methodology’s weakness of relearning the bias through the sensitive group classifier, I am not confident in the validity of this method. In order to prove me otherwise, I encourage the authors to provide the following:

1- Report the difference between the recall performance on different sensitive groups in the test set. For example, for each profession (doctor, for example), I would encourage the author to report the absolute difference between the recall of the Male samples versus Female samples annotated in the test set and then average the metric over all professions. Their method will have a high absolute difference compared to other methods, thus exacerbating the bias.

2- Even if the results above are satisfactory, I would like to see a theoretical or methodological argument proving that the gender classifier will not demonstrate the abovementioned problem.

Furthermore, I encourage the authors to include an experiment where prior work methods like MI-CLIP and FairSample but with explicit bias text in the prompt (e.g., “Male Engineer”) and then perform K/2 retrieval for each gender-based prompt. I expect this will result in a balanced retrieval bag at test time with minimal additional cost.


**Limitations:**

The issue above should be discussed in their limitations sections. I encourage the authors to expand their limitations section and include a more critical discussion of their use of sensitive group classification systems. Overall, I need more evidence about the validity of the contribution. However, I am happy to update my score in case I missed any central point that could be pointed out in the authors’ replies.

---

> ### Author Rebuttal · Authors · 2023-08-09
>
> ### **Response to Reviewer TTMH**
> Thanks for your detailed review and constructive feedback. Before reading our dedicated response to your comments, we kindly suggest you start with our global response, which offers a broad overview and addresses some of the general concerns raised.
> Below is our response to your comments,
>
> #### **"An apparent problem with this approach is that the sensitive group classifier will likely be biased." and "Report the difference between the recall performance on different sensitive groups in the test set..."**
> We acknowledge your concern about the performance and fairness of the demographic group classifier. Regarding the performance concern, since our goal is to mitigate perceivable bias in web image search systems, the group classifier is unlikely to have poor performance because perceivable attributes are in principle easy to classify. Further, in Table 1 (rebuttal), we empirically show that the group classifier works well across different demographic attributes in Occupation 1 & 2 datasets. Please refer to the global response for a more detailed discussion on this issue.
>
> As you requested, we report the absolute difference between the recall performance stratified across different sensitive groups in our test set, as shown in the Table 3 (rebuttal). The results show that PBM achieves the least absolute difference compared to in-processing methods and MI-clip.
>
> #### **"I would like to see a theoretical or methodological argument proving that the gender classifier will not demonstrate the abovementioned problem."**
> It is reasonable to suspect that the demographic group classifier may exacerbate retrieval bias. However, we argue that if we can get a classifier that has good performance when classifying perceivable attributes, thus the PBM classifier can ensure the retrieval bias no worse than retrieving images without any gender or race information. The empirical results of Figure 1 and Figure 2 (rebuttal) support this claim.
>
> ##### **Theoretical analysis of PBM debiasing effect**
> In this analysis, we simplify our focus to a scenario where the only attributes are male and female. The notations below are adopted from our main paper.  To understand the group classifier's influence, we investigate the likelihood of PBM creating a minimally biased image set comprising one male and one female. The probability of successfully selecting a male and female image pair can be expressed as:
>
> $$
> \mathbb{P}[g(v_1) \neq g(v_2)]
> $$
>
> $$
> \qquad = \mathbb{P}[g(v_1) = +1 \wedge \hat{g}(v_1) =1]\mathbb{P}[g(v_2) = -1 \wedge  \hat{g}(v_2) = - 1]
> $$
>
> $$
> \qquad \+ \mathbb{P}[g(v_1) = -1 \wedge  \hat{g}(v_1) = + 1]\mathbb{P}[g(v_2) = + 1 \wedge  \hat{g}(v_2) = - 1] .
> $$
>
> In the above equation, $\mathbb{P}[g(v_1) = +1 \wedge \hat{g}(v_1) = -1]$
> represents the error in misclassifying a man as a woman and
> $\mathbb{P}[g(v_1) = -1 \wedge \hat{g}(v_1) = + 1]$ is the error in misclassifying a woman as a man. We denote them as ϵ1 and ϵ2 as follow,
>
> $$\epsilon_1 =  \mathbb{P}[g(v_1) = + 1 \wedge \hat{g}(v_1) = - 1],
> $$
>
> $$
> \epsilon_2  = \mathbb{P}[g(v_1) = -1 \wedge \hat{g}(v_1) = + 1].
> $$
>
> Then, we can reframe our equation as:,
>
> $$
>  \mathbb{P}[g(v_1) \neq g(v_2)] = (1 - \epsilon_1) (1 - \epsilon_2) + \epsilon_1 \epsilon_2 .
> $$
>
> Suppose the total error of misclassifying one gender as the other is given by,
>
> $$
> \epsilon_1 + \epsilon_2  = \epsilon.
> $$
>
> We can further refine our equation:
>
> $$
> \mathbb{P}[g(v_1) \neq g(v_2)] = 1 - (\epsilon_1 + \epsilon_2) + \frac{1}{2} ((\epsilon_1 + \epsilon_2)^2 - (\epsilon_1 - \epsilon_2)^2)   = 1 - \epsilon + \frac{1}{2}(\epsilon^2 - (\epsilon_1 - \epsilon_2)^2) .
> $$
>
> Subsequently, we have an expression for the likelihood of forming a minimal fair image set in terms of the total error $\epsilon$ and prediction disparity $\epsilon_1 - \epsilon_2 $. From the above equation, the lowest $\mathbb{P}[g(v_1) \neq g(v_2)] $ arises when $|\epsilon_1 - \epsilon_2 | = \epsilon$, suggesting the most significant disparity in predicting male and female with a fixed error $\epsilon$.
>
> Consider the worst case of binary classification is $\epsilon  = 0.5$ and $|\epsilon_1 - \epsilon_2 | \geq 0.5$.
> We have,
> $$\mathbb{P}[g(v_1) \neq g(v_2)] \geq 0.5$$
>
> Remarkably, the above inequality suggests the probability of choosing a fair subset via PBM is no worse than constructing a fair subset by randomly selecting 2 images from a gender-balanced dataset, given equal occurrence for both genders($\mathbb{P}(g(v)=+1)=0.5$ and $\mathbb{P}(g(v) = -1) = 0.5)$.
>
> Therefore, we can conclude that the retrieval bias using PBM can be no worse than choosing samples without any demographic information (which is the goal of in-processing methods), because PBM addresses the dataset bias at test-time. This claim is reinforced by the empirical results of Figure 1 and Figure 2 (rebuttal).
>
> ##### **Empirical analysis of PBM debiasing effect on Occupation 1**
> We show an empirical analysis that is closer to real-world case in Figure 2 (rebuttal). Specifically, we vary the sensitivity ratio between males and females representing the bias by changing the group classification threshold. We show that the resulting retrieval bias is lower than that of the original CLIP for a wide range of sensitivity ratios.
>
> #### **"Furthermore, I encourage the authors to include an experiment where prior work methods like MI-CLIP and"**
> Please see Table 2 (rebuttal) for results of our methods combining with MI-clip and other fine-tuned CLIP.

---

> > ### Comment · Reviewer_TTMH · 2023-08-11
> >
> > Thank you for the detailed response. Indeed, while the gender classifier performance seems to have an impact on the recall per Figure 1 (rebuttal), the performance on the occupation dataset seems adequate to ensure good performance (Table 3). I encourage the authors to update the paper writing and include these new results in order to ensure no confusion for future readers. Eitherway, given these results, I am more confident about the ability of a reasonably accurate gender classifier to ensure adequate recall performance for PBM. It is also nice to see that combining PBM with prior work Fairness methods further improves these methods and sometimes outperforms PBM; this makes sense, given that the features are more robust.
> >
> > Overall, given the novelty of the problem definition and the adequate initial proposed solution, I upgraded my review to Weak Accept.

---

> > > ### Author Response · Authors · 2023-08-12
> > >
> > > Thank you for your feedback and the decision to upgrade the review score for our manuscript. We genuinely appreciate your recognition and trust in our work. We will incorporate your suggestions in the revision of our paper.

---

### Official Review · Reviewer_KCNe · 2023-07-06

**Soundness:** 3 good
**Presentation:** 3 good
**Contribution:** 3 good
**Rating:** 6
**Confidence:** 4

**Summary:**

This paper proposes a post-hoc fairness method for text-to-image retrieval tasks. Instead of pre-processing the dataset or in-processing via fairness-derived regularization, this paper focuses on the post-processing of the retrieved items. The proposed post-hoc bias mitigation (PBM) utilizes a pre-trained bias estimator, e.g., a gender classifier. Using the gender classifier, PBM first classifies the gallery set (or test set) into male / female / NA sets. Then, it samples the final retrieval items from each set by balancing each gender for the final retrieval results. The experimental results show that the proposed PBM can show great bias mitigation while showing small performance drops for the target task.

**Strengths:**

- This paper addresses an important problem in vision-language tasks, especially for text-to-image retrieval tasks.
- The analysis of the source of the bias ("Two Types of Bias") is insightful and well-described.
- The proposed method is easy-to-implement and practical. Moreover, as this method is a post-processing method that does not require a re-training of the model, this method can be readily applied to any text-to-image retrieval method.
- Although the proposed method highly relies on the performance of the pre-trained gender classifier, this paper extensively studied many possible alternatives of the gender classifier, e.g., zero-shot embedding or zero-shot prompt. It is also pleasurable to see the "oracle", or the upper bound of the proposed method, by employing ground-truth gender labels. It shows the advantages and disadvantages of the proposed method more clearly.
- This paper is easy-to-follow

**Weaknesses:**

- The proposed method needs a pre-trained gender (bias) classifier. As shown in the experiments, the performances of PBM heavily rely on the quality of the gender (bias) classifier. Although this paper shows a generalizable approach using zero-shot prompts, I think it is an unavoidable weakness of this paper.
- Compared to in-processing methods, PBM shows worse target task performances. I think it depends on the target objective of the actual user: if one needs a highly performed text-to-image retrieval system (i.e., with high recall) while allowing somewhat moderate discrimination, in-processing methods would be better. On the other hand, if one needs a strictly fair text-to-image retrieval system while sacrificing recall performances, the proposed PBM would be better. I would expect related discussions in the future revision, if possible.
- As another fundamental limitation, PBM cannot improve the original target task (e.g., Recall@K) because it is a post-processing method.
- The notion of fairness can be written in multiple ways. For example, the target fairness problem in this paper is similar to "demographic parity (DP)" problem, rather than "equalized odds (EO)". As far as the reviewer understood, mutual information minimization (MI) is more focusing on EO, while the proposed method is focusing on DP. In my opinion, both EO and DP are valuable problems, while a user has to choose a better fairness notion by considering the user's application. I think the problem raised by this paper is very important, especially considering that it is a retrieval task, not a classification task, but in my opinion, it would be helpful for the readers to discuss the target fairness scenario and the benefit of the target fairness scenario.

**Questions:**

As my previous comment, I would like to expect more discussions related to the notion of fairness chosen in this paper. In my opinion, the notion of fairness can vary by situation.

**Limitations:**

I don't think this paper has a potential negative societal impact.

As my previous comment, the proposed method has fundamental performance limitations caused by (1) the need of an accurate gender classifier (2) the nature of the post-processing method (PBM does not update the original model, therefore, PBM cannot improve R@K).

I think this paper is a generally okay paper, while its strengths slightly outweigh its flaws. In my opinion, this paper would become stronger if discussions of the target fairness scenario and other fairness scenarios when the listed limitation would be / not be critical.

---

> ### Author Rebuttal · Authors · 2023-08-09
>
> ### **Response to Reviewer KCNe**
> Thanks for your detailed review and constructive feedback. Before reading our dedicated response to your comments, we kindly suggest you start with our global response, which offers a broad overview and addresses some of the general concerns raised.
> Below is our response to your comments,
>
> #### **"The proposed method needs a pre-trained gender (bias) classifier. ..."**
> We agree with your observation that our PBM method's performance is contingent upon the quality of the demographic group classifier. In our global response, we argue that our goal is to mitigate perceivable bias in web image search system, which alleviates the chances of the group classifier for having poor performance because perceivable attributes are in principle easy to classify. Further, in the Table 1 (rebuttal), we empirically show that the group classifier works well across gender and race demographic attributes in Occupation 1 and 2 datasets. We have not tested our approach with demographic attributes that are difficult to predict, thus this will be discussed and emphasized as a limitation in the revision.
>
> #### **"Compared to in-processing methods, PBM shows worse target task performances. ..."**
> We acknowledge that our PBM eliminates bias at the cost of causing retrieval performance drops that are more severe than other in-processing methods. However, we provide a simple bias/recall trade-off strategy by introducing $\theta$, a parameter controlling the likelihood of obtaining a fair set. As a result, $\theta$ can be used to prioritize recall as needed. We present the results of bias-recall trade-off curve in Figure 4 (paper). We will include a dedicated section in revision to discuss trade-off strategy, including how to choose $\theta$ under different application context and pseudo-codes of PBM trade-off algorithm, to avoid confusions.
>
> #### **"As another fundamental limitation, PBM cannot improve target task..."**
> We acknowledge that our PBM is unable to improve the target task performance with a fixed underlying image retrieval model. However, one can fine-tune the CLIP on the target task dataset. In Table 2 (rebuttal), we show the results of applying PBM on a fine-tuned or modified CLIP, including MI-clip, Adversarial Learning and Debias Prompt. The PBM - Debias Prompt slightly improve the recall since the Debias Prompt model is fine-tuned on additional data. We will emphasize this point in the limitation section of the revision.
>
> #### **”The notion of fairness can be written in multiple ways. ...“**
> We acknowledge that fairness can be defined in a multitude of ways under different contexts. This will be emphasized in the revision. In our paper, fairness is defined as "equal representation", i.e., that the output retrieval bag should include an equal number of samples from each demographic group. We write down the mathematical definition of "equal representation" used in our paper as follows,
>
> $$
> \mathbb{E}_{V_c \sim P}[\mathbb{E}_v[\mathbb{1}(g(v)=+1)]]
> $$
>
> $$
> \qquad = \mathbb{E}_{V_c \sim P}[\mathbb{E}_v[\mathbb{1}(g(v)=-1)]]
> $$
>
> , where P is the image set distribution. Other notations follow from the paper.
>
> Our definition of equal representation is equivalent to the definition 3 of "Equal opportunity for ranking distributions" in [1] when all ranking position biases are equal.
>
> Further, we would like to clarify that the fairness problems of "EO" and "DP" are for classification tasks. In retrieval tasks, we use "equal representation" as the goal of the fairness. Since "EO" and "DP" are defined in a classification setting, they cannot be directly applied to our retrieval setting, where the output is a bag of images. However, one can extend the notion of "EO" and "DP" to other tasks, similar to [1]. This will discussed in the introduction of the revised paper.
>
> [1] Singh, Ashudeep, and Thorsten Joachims. "Equality of opportunity in rankings." Workshop on Prioritizing Online Content (WPOC) at NIPS. Vol. 31. 2017

---

> > ### Comment · Reviewer_KCNe · 2023-08-16
> >
> > Thanks for the authors answering my questions.
> >
> > As my initial concerns are centered around its presentations that can be enhanced by minor modification, I will keep my rating as "weak accept". The listed revision plan looks great to me.
> >
> > Also, thanks for pointing out that DP and EO are defined for classification tasks. I totally agree with the response, but still, there are many readers who are not unfamiliar with algorithmic fairness. I would like to encourage the authors to emphasize the significance of the proposed fairness scenario in the revision.

---

> > > ### Author Response · Authors · 2023-08-17
> > >
> > > Thank you for acknowledging our response and the proposed revision plan. In our revision, we will place greater emphasis on the proposed fairness scenario. We appreciate your constructive feedback and we are committed to enhancing the quality of our work based on your input.

---

### Author Rebuttal · Authors · 2023-08-09

We sincerely appreciate your valuable feedback. We are grateful for the constructive comments and suggestions, which provided us with insights to improve the quality and clarity of our work. Below, we address the common concerns raised by the reviewers.

### **Concerns on the performance of group classifiers**
We acknowledge that the performance of the proposed PBM is affected by the accuracy and bias of the demographic group classifier, which will be emphasized in the revision. Regarding the former, in Figure 1 (rebuttal) we demonstrate the relationship between the performance of group classifier and the resulting retrieval bias by artificially introducing noise to the demographic group (logit) predictions via Gaussian noise with standard deviation varying from 0 (no noise) to 1. These results illustrate that better group classifier performance yields lower bias, that bias converges to that of the original CLIP as the group classifier gets worse, and importantly, that the bias after PBM will be no worse than that of the original CLIP.

Further, Table 1 (rebuttal) shows the individual demographic group sensitivities under three different scenarios, from which we can see that the group classifier is i) able to achieve good classification sensitivity (no lower than 0.81 and 0.90 in average), likely because demographic image attributes (gender and skin tone) are typically captured in images, and ii) that different scenarios exhibit different degrees of bias as measured by the group sensitivity ratio, which must be close to 1 for the model to be unbiased. Note that bias in (binary) classification can be measured in many ways other than sensitivity ratios. This will be acknowledged in the revision.

For the concerns about the bias of the group classifier, we acknowledge that there are sensitivity discrepancies between different attributes, as shown in Table 1 (rebuttal). To further investigate the influence of the group classifier bias on the PBM results, we present the retrieval bias as a function of the sensitivity ratio in Figure 2 (rebuttal). This is achieved by changing the group classification threshold from 0 (max male sensitivity) to 1 (minimum male sensitivity). As pointed out by the reviewers, group classifier bias affects retrieval bias, however, only severely for more extreme sensitivity ratios, which is fortunately not the case in our results as shown in Table 1 (rebuttal).

To show the impact of group classification calibration on PBM, we leverage a simple post-hoc calibration method, histogram binning [4], on the worst performing method in Table 1 (rebuttal). We calibrate the group classifier on Occupation 2 - Gender and then test it on Occupation 1 - Gender. The calibrated group classifier achieves a sensitivity ratio of 1.01 (93.5\% male sensitivity and 92.4\% female sensitivity). The resulting AbsBias@100 using the calibrated group classifier is 0.1848 and the Recall@100 is 51.3\% on Occupation 1 - Gender, which results in worse retrieval bias and better recall relative to the uncalibrated group classifier in Table 1 (paper). We believe this is because the retrieval dataset (Occupation 1 - Gender) is imbalanced, which may have a negative impact on the resulting AbsBias@100 after PBM. This will be highlighted as a potential limitation of PBM.

Finally, we emphasize that the main contribution of our work is to identify the occurrence of test-time bias, which significantly influences image retrieval results and manifests as a result of the need for selecting samples from a test pool (the retrieval set), which are likely to be correlated, thus prone to introduce bias, more so if the test pool itself is biased. Consequently, we seek to analyze and raise awareness of the test-time bias issue, which is critical in image retrieval scenarios. Our work can be seen as a first step that can be greatly improved with additional research.

[1] Pleiss, Geoff, et al. "On fairness and calibration." NeurIPS 30 (2017).

[2] Zafar, Muhammad Bilal, et al. "Fairness constraints: A flexible approach for fair classification." JMLR (2019).

[3] Celis, L. Elisa, et al. "Classification with fairness constraints: A meta-algorithm with provable guarantees." ACM FAccT (2019).

[4] Zadrozny, Bianca, and Charles Elkan. "Obtaining calibrated probability estimates from decision trees and naive bayesian classifiers." ICML (2001).

### **Concerns about the recall performance drop using PBM**
For the COCO and Flickr30k benchmarks, the retrieval pools for each query are very limited. When we apply PBM, the recall performance suffers because there are usually not enough samples in the minor demographic groups to satisfy the PBM constraint. For instance, there are queries mostly describing one gender like "A person wearing a dress..." in COCO, which make the debiasing problem very hard. Further, the performance drop in Occupation 2 race classification is much larger because there are simply no sufficient images for minor skin-tones in some occupations. For example, occupations "editor" and "photographer" have only 2 and 5 images annotated as "dark skin", respectively. This is a sample size limitation that will be emphasized in the revision.

There is a fundamental trade-off between bias and recall, and Table 1 and 2 (paper) only show results where we prioritize low bias rather than maintaining high recalls. For users who seek prioritizing retrieval performance, our method can be easily adjusted to prefer recall over bias, as illustrated in Figure 4 (paper) and discussed in the ``Bias-performance trade-off analysis'' paragraph in Section 4. Specifically, Figure 4 shows that for a fixed recall value, our method has significantly lower bias than other methods. This supports that our method is able to reduce bias while maintaining comparable recalls. We agree that reading the results in Table 1 and 2 alone could be misleading, thus we will clarify the results in Figure 4 in the revision.

---

### Decision · Program_Chairs · 2023-09-21

**Decision:**

Accept (poster)

**Comment:**

All reviewers are supportive of this paper post-rebuttal. The identified weaknesses, like dependence on pre-trained gender-classifier, validity of method, and several limitations, have been adequately addressed in the rebuttal and should be incorporated in the cam-ready to further strengthen the work.